# The computation of directional selectivity in the *Drosophila* OFF motion pathway

**Eyal Gruntman, Sandro Romani*, Michael B Reiser***

Janelia Research Campus, Howard Hughes Medical Institute, Ashburn, United States

**Abstract** In flies, the direction of moving ON and OFF features is computed separately. T4 (ON) and T5 (OFF) are the first neurons in their respective pathways to extract a directionally selective response from their non-selective inputs. Our recent study of T4 found that the integration of offset depolarizing and hyperpolarizing inputs is critical for the generation of directional selectivity. However, T5s lack small-field inhibitory inputs, suggesting they may use a different mechanism. Here we used whole-cell recordings of T5 neurons and found a similar receptive field structure: fast depolarization and persistent, spatially offset hyperpolarization. By assaying pairwise interactions of local stimulation across the receptive field, we found no amplifying responses, only suppressive responses to the non-preferred motion direction. We then evaluated passive, biophysical models and found that a model using direct inhibition, but not the removal of excitation, can accurately predict T5 responses to a range of moving stimuli.

**\*For correspondence:**
romanis@janelia.hhmi.org (SR);
reiserm@janelia.hhmi.org (MBR)

**Competing interests:** The authors declare that no competing interests exist.

## Introduction

Visual motion is not directly measured, rather it is computed by neuronal circuits downstream of photoreceptors. This computation is fundamental to the extraction of many visual features, is a local operation implemented by a small circuit, and is simple enough to be approximated by compact algorithms. Although several different models have been proposed for generating directionally selective responses across species, they all share three common elements: (1) spatially offset inputs with a (2) temporal asymmetry between them, that are (3) non-linearly combined. These models generate Directional Selectivity (DS) by enhancing responses in the preferred direction (seen in flies: *Fisher et al., 2015*; *Salazar-Gatzimas et al., 2016*), suppressing responses in the null, or non-preferred direction, or by a combination of both (also from flies; *Haag et al., 2016*; *Leong et al., 2016*; *Strother et al., 2017*, reviewed in *Yang and Clandinin, 2018*). In flies, local luminance increments (ON) and decrements (OFF) are processed by largely separate circuits, with motion being computed separately in each pathway (*Behnia et al., 2014*; *Clark et al., 2011*; *Franceschini et al., 1989*; *Joesch et al., 2010*; *Silies et al., 2013*; *Strother et al., 2014*; *Takemura et al., 2013*). The first neurons to generate directionally selective responses are the T4 cells of the ON pathway and the T5 cells of the OFF pathway (*Maisak et al., 2013*; *Serbe et al., 2016*; *Strother et al., 2017*) (*Figure 1A*).

T4 neurons use both local excitatory and inhibitory inputs to generate directionally selective responses (*Gruntman et al., 2018*; *Haag et al., 2017*; *Strother et al., 2017*; *Takemura et al., 2017*). A recent connectomics study has characterized all the columnar inputs to T5 neurons (*Shinomiya et al., 2019*) and a functional imaging study revealed they all depolarize in response to OFF stimuli (*Serbe et al., 2016*). However, transcriptional profiling of these neuron types shows that these inputs are all cholinergic, and therefore unlikely to provide local inhibitory input (*Davis et al., 2018*). Does the OFF pathway use a different algorithm to compute motion? To address this question, we performed whole-cell recordings of T5 neurons while presenting visual stimulation. First, we used single bar flashes to map the first-order Receptive Field (RF). These stimuli, which do not

contain any motion information, allowed us to map the spatial distribution of depolarizing and hyperpolarizing inputs and revealed that T5 neurons receive local hyperpolarizing inputs. Next, we used many variants of pairwise bar flashes, an elementary motion stimulus, and consistently found only a single mechanism responsible for DS. Next, we constructed a conductance-based model for a T5 cell and used it to predict responses to an array of visual stimuli. The model is comprised of fast excitation and slow inhibition spatially offset to the trailing side of the RF (*Gruntman et al., 2018*). We show that our model, constructed only from the first-order RF responses that contain no motion information, predicted responses to dynamic stimuli, such as moving bars and drifting gratings. Finally, we compared the behavior of our model with a model based on the removal of excitation (rather than direct inhibition). We show why a model based on offset excitation and inhibition is more directionally selective and is also better at recapitulating T5 responses.

## Results

### Whole-cell recordings of T5 neurons show small-field, directionally selective responses

We measured visual responses of T5 neurons by targeting in-vivo whole-cell electrophysiological recordings to their GFP-labelled somata and presenting stimuli on a hemi-cylindrical LED display (*Figure 1B*). We confirmed the identity of the labeled neurons as T5 cells by recording reliable depolarizations in response to small OFF flashing squares (~11˚×11˚; pixels turned off from an intermediate-intensity background). These flashing squares were also used to localize the RF center (*Figure 1C*) in a step-wise process that mapped the maximal response position for each recorded neuron by probing smaller areas at higher resolution until the peak response was localized to a single pixel (~2˚×2˚) of the display. Having localized the RF center, we then evaluated DS for each neuron by measuring responses to bars of three widths moving through the RF center along eight directions. The bars moved at the speed (28˚/s) that produced the largest directionally selective responses in T4 neurons (*Gruntman et al., 2018*). Each of the 17 T5 neurons we recorded showed clear preferred and null directions (PD and ND) of motion, with relatively wide tuning (i.e. similar magnitude responses in the directions ± 45˚ away from the PD; individual neuron example *Figure 1D*; population responses *Figure 1E*). The PD response shows a clear hyperpolarization following the depolarizing peak, and the ND response shows a dip preceding the depolarization peak. We quantified these responses with a Directional Selectivity Index (DSI = $[PD_{max} - ND_{max}]/PD_{max}$), and found that this measure is significantly different from zero for the three widths tested (*Figure 1E*). We note that the smallest bar presented (1-pixel wide,~2˚) generated a clear directionally selective response but did not induce the prominent hyperpolarization observed for the wider bars. We further note that although wider bars evoked stronger responses, they did not increase the DSI (*Figure 1E*), because these stronger stimuli also evoked a corresponding increase in the ND response magnitude. The response dynamics and the width of the directional tuning are similar to our T4 responses to moving bars (Figure 1 of *Gruntman et al., 2018*).

### T5 receptive field is comprised of spatially offset depolarization and hyperpolarization

Classic studies of mammalian directionally selective neurons used decomposable motion stimuli to map spatial responses to the individual components (the 'first-order RF'), and to their pairwise interactions (the 'second-order RF'). This procedure was important in ruling out competing models for generating DS (*Barlow and Levick, 1965*; *Emerson et al., 1992*; *Emerson et al., 1987*; *Jagadeesh et al., 1993*). In our previous work (*Gruntman et al., 2018*), we adapted this approach to T4 neurons, and found that a fine-scale characterization of the neuron's first-order RF was incompatible with the Hassenstein-Reichardt model (*Hassenstein and Reichardt, 1956*), the predominant model for computing DS in insects. We wondered whether the differences between the signs of T4's and T5's columnar inputs would also produce functional changes in the first-order RF. We therefore used on-line stimulus generation to map the spatial and temporal properties of T5 receptive fields using flashing bars presented along the identified PD-ND axis of each cell (*Figure 1*). We used bars of width 1, 2, and 4 pixels (corresponding to 2.25˚, 4.5˚, and 9˚ of visual angle), and our RF maps are plotted (*Figure 2*) along a stimulus position axis, where a unit change in position is equivalent to a

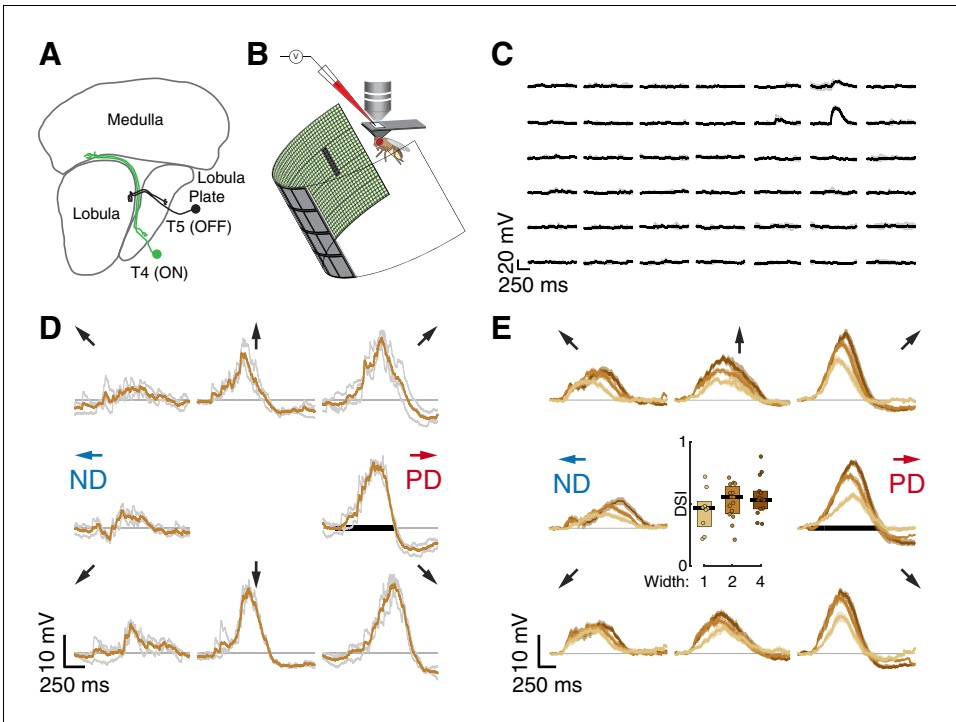

**Figure 1.** Whole-cell recordings of T5 neurons show small-field, directionally selective responses. (**A**) Schematic of the *Drosophila* visual system with an example T4 (ON) and T5 (OFF) neuron. (**B**) Schematic of experimental setup. Whole-cell recordings were targeted to soma of GFP-labeled T5 neurons. (**C**) Responses to 200 ms OFF square flashes (~11°×11°) from an example cell. Each subplot shows the response from a different location on the LED display, which subtended 216° (azimuth) ×72° (elevation) of the visual space. Individual repeats in gray (n = 3 trials), mean in black. (**D**) Responses from the same cell in (**C**) to a 2-pixel wide dark bar (2 × 9 LEDs,~4.5°×20.25°) moving in eight directions at 28°/s (80 ms/pixel) through the center of the receptive field. Repeats in gray (*n* = 3 trials), mean in brown. Black horizontal bar indicates the stimulus duration. PD indicates the Preferred Direction and ND indicates the Null Direction. (**E**) Baseline-subtracted responses (*n* = 17 cells) to a moving bar of width 1, 2, and 4 pixels (2.25°, 4.5°, and 9°), aligned to the PD of each cell (mean ± SEM). Arrows represent the direction of stimulus motion. Black horizontal bar indicates stimulus presentation. Inset: DSI = [$PD_{max}$ − $ND_{max}$]/$PD_{max}$ for moving bar responses (*n* = 9, 15, 14 cells).

1-pixel movement of a width 1 bar (see *Figure 2—figure supplement 1* for mapping onto visual angle for T5 neurons with PDs aligned to diagonal and cardinal directions on our display). Although these single position bar flashes lack directional information, the temporal and spatial structure of T5 responses to them make up the first-order RF that is used for all further comparisons, since more complex stimuli can be comprised of different concatenations of single position flashes.

Aligning the responses from all the cells based on the position of the peak depolarization (*Figure 2A*) reveals T5's (first-order) RF structure: (1) Depolarizing responses dominate the center, with increased response magnitude for stronger stimuli (wider bars or flashes of longer duration). This increase is not linear. For example, at certain positions responses cannot be detected for weak stimuli but clear responses can be seen for stronger stimuli (e.g. compare bar width 1 at position 5, gray vs. brown traces, downwards arrowhead). (2) Inputs along the PD-ND axis are spatially asymmetric around the RF center. On the leading side of the RF the responses reflect depolarizing input (*Figure 2A*, green positions), while on the trailing side, responses show a mixture of depolarization and hyperpolarization (*Figure 2A*, pink positions). This general structure is similar to T5 measurements in a recent paper using voltage imaging (*Wienecke et al., 2018*). Although stronger stimuli (wider bars or longer duration flashes) induce a stronger hyperpolarizing component in the response, this effect is also non-linear. For example, average traces in position 3 (bars of width 2) show no hyperpolarization for a 40 ms flash (gray trace), but a prominent hyperpolarization when the flash duration is 160 ms (brown trace, *Figure 2A*, upwards arrowhead).

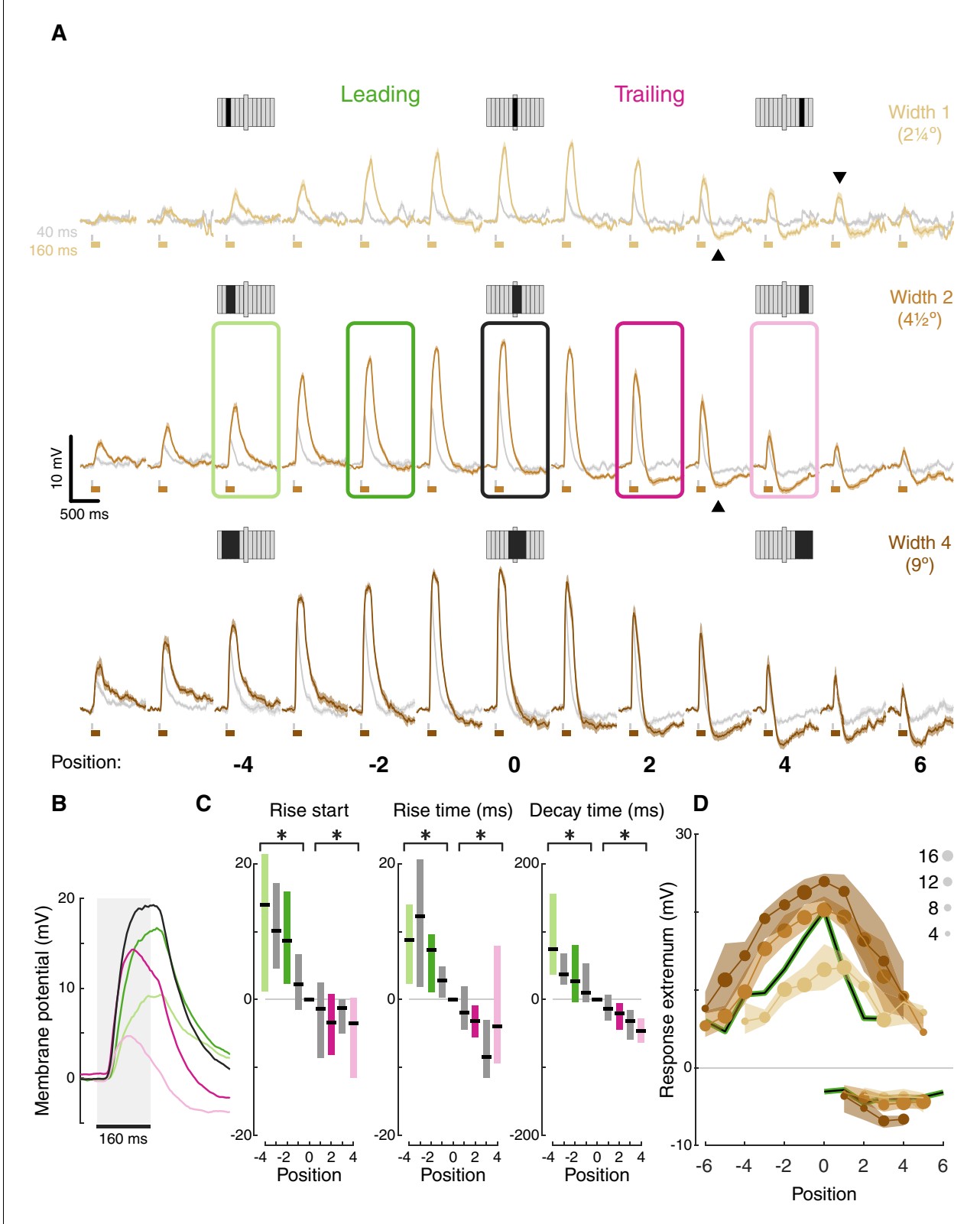

**Figure 2.** T5 receptive field is comprised of spatially offset depolarization and hyperpolarization. (**A**) Averaged, baseline-subtracted responses (mean ± SEM) to bar flash stimulus at the indicated positions (numbered below, examples schematized above) along the PD–ND axis of each cell (*n* = 17 cells) aligned to the central depolarizing position (=0, see Materials and methods for details). Responses to 40 ms flashes in gray; responses to 160 ms flashes are colored. Elongated bar in each stimulus schematic marks center position. Downwards arrowhead: example non-linearity in responses

*Figure 2 continued on next page*

*Figure 2 continued*

to stimuli of different durations. Upwards arrowhead: example non-linearity in hyperpolarization in response to stimuli of different durations. (B) Mean responses from indicated positions in (A) of width-2 bar flashes, aligned to stimulus presentation (gray rectangle). (C) Response rise start time (time to reach 10% of max), rise time (10%–50%), and decay time (80%–20%) for positions surrounding the receptive field center (calculated for 160 ms flashes of width-2 bars). Results presented as differences from the central position (* indicates significantly above/below zero for pooled positions from the leading/trailing side respectively, two-tailed t-test, p<0.05; n = 17 cells). (D) Maximum depolarizing and hyperpolarizing responses at each stimulus position for 160 ms bar flashes of all three widths. Dots correspond to median response (size indicates the number of cells for each position), shaded regions demarcate upper and lower quartiles. Superimposed black line represents the median T4 results for width 1 ON bars (*Gruntman et al., 2018*). See also *Figure 2—figure supplement 1* for the T5 receptive field corrected for visual angle.

The online version of this article includes the following figure supplement(s) for figure 2:

**Figure supplement 1.** T5 receptive field comparison between cells aligned to cardinal and diagonal preferred directions, corrected for approximate visual angle.

The unipolar structure of *Drosophila* neurons precludes the use of voltage clamp to separate the inhibitory and excitatory conductances in the responses. However, a closer examination of the response dynamics from the leading and trailing side of the RF reveals a temporal interaction between depolarization and hyperpolarization. Long duration responses on the leading side show a plateau-like depolarization peak, while responses on the trailing side show a peak that rapidly decays (*Figure 2B*, pink vs. green responses). These decaying response peaks likely arise from a 'competition' between fast-rising, fast-decaying depolarizing inputs and slow-rising, slow-decaying hyperpolarizing inputs. At stimulus onset, the fast depolarization is dominant. As the slower hyperpolarization increases with time, so does the depolarization decay. At stimulus offset, depolarization decays rapidly, while hyperpolarization persists beyond the stimulus offset (*Figure 2B*).

Spatially asymmetric hyperpolarization also induces faster decaying responses on the RF's trailing side (*Figure 2B* pink vs. green, quantified in *Figure 2C*). In T4 neurons, we showed that the asymmetric, slow-decaying hyperpolarization is the main contributor to DS generation, since ND movement first activates the faster-decaying depolarizing response components, which in turn lead to less efficient summation and smaller net depolarization (*Gruntman et al., 2018*). Therefore, the faster decay times on the trailing side of T5 neurons were also the likely result of hyperpolarization temporally sharpening responses. Unlike in T4, trailing-side responses started sooner and rose faster than leading-side responses (*Figure 2C*; two-tailed t-tests, p<0.05 for all conditions). It is unclear whether these differences in rise start and rise time play a role in generating directionally selective responses, or whether they simply reflect a wider range of spatially offset excitatory input types in T5 cells (*Shinomiya et al., 2019*).

Plotting the peak depolarization and hyperpolarization for each cell at each position for long duration flashes of all widths illustrates the RF structure described above: depolarizing component decaying symmetrically from the center, and an asymmetric hyperpolarizing component on the trailing side (*Figure 2D*). This structure is comparable to the RF structure we measured for T4 cells (*Figure 2D*, black curves). Note that the T4 RF was mapped using responses to a bar of width 1, suggesting a potential size-sensitivity difference between the ON and the OFF pathways (consistent with *Haag et al., 2017*). The high degree of similarity in the first-order RF between T4 and T5 cells suggests that both cell types receive similar (functional) inputs. But do they use a similar DS mechanism? To test this directly we next characterized T5 responses to elementary motion stimuli.

## T5 neurons generate directional selectivity using ND suppression

Sequential presentation of two adjacent bar flashes, sometimes referred to as two-step apparent motion, is a common stimulus used to map the second-order RF of directionally selective cells by comparing the superposition (summation shifted in time) of the first-order responses with the responses to the sequential, two position stimulation (*Figure 3A*). This comparison is a direct test for the relative contribution of either PD enhancement or ND suppression in generating the directionally selective response (*Figure 3A*). To illustrate the general response properties, we focus first on responses to width two bars that were each presented for 160 ms (*Figure 3B,C*). Responses on the leading side of the RF, up to and including the center (position 0), did not exhibit DS. Since both PD and ND combinations evoke a similar maximal response (e.g. −3 to −1 versus −1 to −3, horizontal lines) the neurons are effectively 'motion blind' in this region of the RF. Responses to

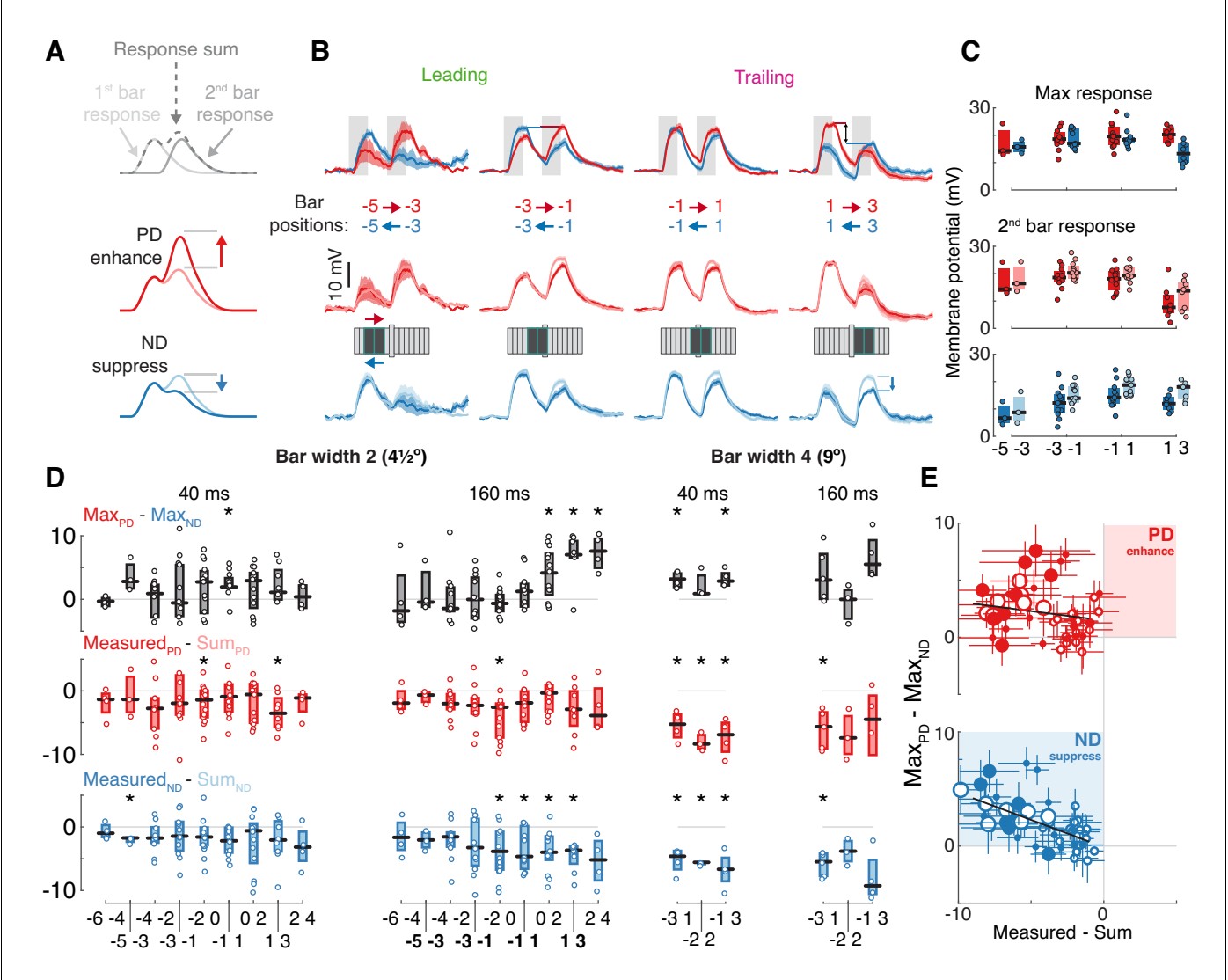

**Figure 3.** T5 neurons generate directional selectivity using ND suppression. (**A**) Schematized responses to the elementary motion stimulus of sequential bar pair flashes. Response could be the sum of the responses to the individual flashes (top), could show preferred direction, PD, enhancement (middle) or null direction, ND, suppression (bottom). (**B**) Baseline-subtracted responses (mean ± SEM) to bar pair combinations presented at different positions along the PD–ND axis (n = 3,11,11,8 cells). Top: responses to PD and ND bar pairs. Middle: responses to PD bar pairs (same as top) with the temporally aligned sum of the responses to the component bar flashes in pink. Bottom: responses to ND bar pairs (same as top), sum of responses to component bar flashes in cyan. Stimulus presentation interval indicated with gray rectangles. Stimulus schematic shows positions of both bars (0 position depicted by elongated bar). (**C**) Boxplot summary of response maxima for (**B**) (same color conventions). Top: response maxima, Middle and Bottom: second bar maxima for measured and summed responses. (**D**) Boxplot summary of response maxima differences for all (non-overlapping) bar pair stimuli presented. Top: difference between PD and ND, positive value indicates a directionally selective response. Middle: difference between second bar response maxima for measured and summed PD responses. Positive value indicates PD enhancement. Bottom: same as middle, but for ND. Negative value indicates ND suppression (* indicates mean significantly differs from zero, unpaired t-test corrected for multiple comparisons by controlling for the false discovery rate with q = 0.075). Boldface positions are presented in (**B**) and (**C**). (**E**) Comparison of directional selectivity versus linearity of response for all presented bar pair combinations, including data from (**D**) and responses to overlapping positions (see Materials and methods). Each dot corresponds to the mean (of n ≥ 3 cells) response differences for each position pair (± SEM). Marker size indicates bar width (small for 2, large for 4-pixel wide), marker fill indicates duration (empty 40 ms, filled 160 ms). Results of linear regression in black (non-significant slope for Measured$_{PD}$ - Sum$_{PD}$ vs. Max$_{PD}$ - Max$_{ND}$, [−0.402, 0.073], 95% confidence interval; significant slope for Measured$_{ND}$ - Sum$_{ND}$ vs. Max$_{PD}$ - Max$_{ND}$, [−0.641,−0.301], 95% confidence interval). See also **Figure 3—figure supplement 2** for more details of bar width four responses.

The online version of this article includes the following figure supplement(s) for figure 3:

**Figure supplement 1.** Evidence for ND suppression is robust to measurement type.

**Figure supplement 2.** Apparent motion responses show evidence only for ND suppression even for larger stimuli.

combinations that included the trailing side of the RF, where hyperpolarization was detected (*Figure 2D*), exhibited a larger maximal response for the PD combination, and therefore exhibited DS (e.g. *Figure 3B* 1 to 3 in red versus 3 to 1 in blue, horizontal lines).

To further explore the mechanism underlying DS we compare each two bar stimulus response (*Figure 3B* middle and bottom, darker traces) to the superposition of the component responses (*Figure 3B* middle and bottom, lighter traces). In the leading side of the RF, where T5 is 'motion blind,' we find that the sequential responses are well approximated by the sum of the individual responses. However, in the trailing side of the RF, where T5 generates directionally selective responses, this comparison reveals suppression of ND responses (blue arrow), as can be seen in the comparison between responses to the second bar (whose location provides the directional component of the stimulus pair; *Figure 3C*). We note that there are no conditions where the summed responses are smaller than the measured responses, regardless of whether the comparison is made using response maximum or mean (*Figure 3—figure supplement 1*). In other words, no combinations of 4.5° bars showed PD enhancement. A recent paper using calcium imaging reported PD enhancement in both T4 and T5 cells, but only for stimuli above a certain size (>6° for T5s; *Haag et al., 2017*). However, our findings hold for all of the apparent motion conditions we tested: fast (40 ms) and slow (160 ms), bars of width 2 (4.5°) and width 4 (9°), and sequential positions that were either adjacent or overlapping (*Figure 3D,E* and *Figure 3—figure supplement 2*). Although PD responses exhibited suppression (*Figure 3D*, middle), these stimuli still evoked directionally selective responses in trailing side positions due to the asymmetric structure of the RF and the even larger suppression of ND stimulation (*Figure 3D*, top).

To summarize the responses to all of these different pairwise combinations we plot the difference between the peak responses of each PD and ND sequence pair (a measure of DS, with a positive value indicating PD preference) against the difference between the measured response and the sum of the component responses (indicating suppression when negative, enhancement when positive). Although we used an expansive stimulus set, we found numerous combinations showing ND suppression and not a single condition showing an enhanced PD response (this is again valid if the comparison is based on mean responses, see *Figure 3—figure supplement 1D*). Furthermore, we found a significant correlation between the magnitude of ND suppression and the magnitude of the directionally selective response (more suppression correlates with more DS), while no such correlation was found for PD responses (*Figure 3E*; $\text{Measured}_{ND} - \text{Sum}_{ND}$ vs. $\text{Max}_{PD} - \text{Max}_{ND}$ $R^2$: 0.396, slope 95% confidence interval: [−0.64,–0.31]; $\text{Measured}_{PD} - \text{Sum}_{PD}$ vs. $\text{Max}_{PD} - \text{Max}_{ND}$ $R^2$: 0.02, slope 95% confidence interval: [−0.402, 0.073]). Taken together, these data provide strong evidence for the suppression of ND motion as the only mechanism through which DS is generated in the membrane potential of T5 neurons.

## A conductance-based model quantitatively predicts directionally selective responses

We constructed a conductance-based neuronal model to test our intuitive proposal regarding the generation of directionally selective responses in T5 cells. The model—henceforth referred to as the EI model—includes an excitatory and an inhibitory conductance that are combined using a biophysically inspired non-linearity (*Figure 4A*). The model, which is slightly simplified from our previous T4 model (*Gruntman et al., 2018*), is parameterized with spatial and temporal filters for each conductance (*Supplementary file 1* and Materials and methods). Based on our previous T4 modeling (*Gruntman et al., 2018*), we hypothesized that the passive integration of these conductances would be sufficient to explain T5's response dynamics. Our objective in constructing the EI model was not only to generate DS responses, but to explore whether the information contained in the responses to non-moving stimuli is sufficient to predict the neuron's responses to more complex stimuli, such as drifting gratings.

We optimized model parameters using an iterative, non-linear, least squares procedure to minimize the difference between the numerical simulation and measured responses for width 2 bar flashes (*Figure 4B*). Importantly, since we optimized only based on responses to static flashing bars, the model parameters were not influenced by any motion-related responses. The model's responses to any stimulus are then simply the result of passive integration of excitatory and inhibitory conductances injected with a temporal pattern determined by the spatial and temporal structure of the stimulus (see Materials and methods). Since our aim was to predict responses to more complex

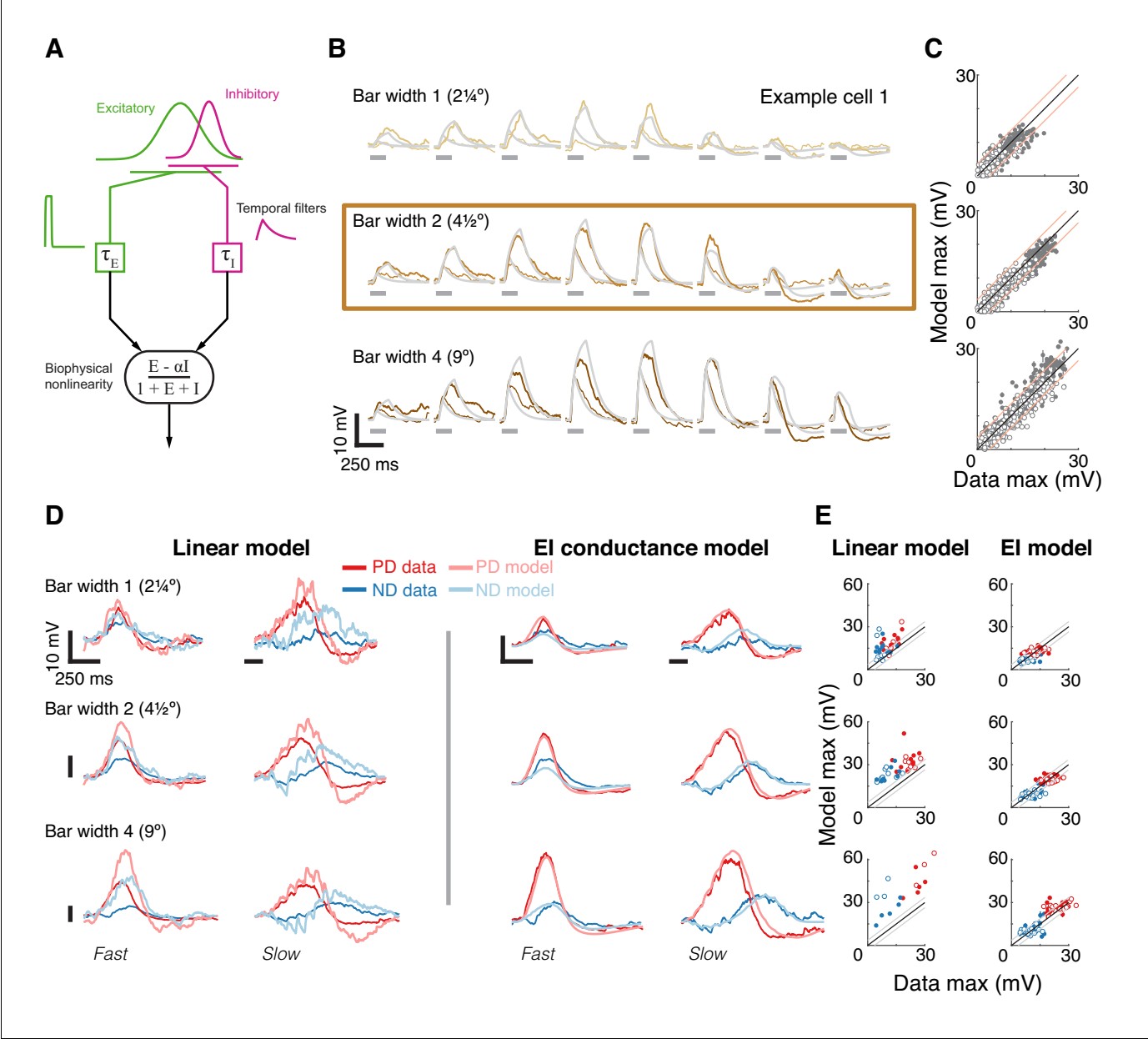

**Figure 4.** A conductance-based model quantitatively predicts directionally selective responses. (A) T5 EI model schematic with fast spatially symmetric excitation and slow, trailing-side asymmetric inhibition. (B) Mean measured responses to single bar flashes of 3 widths and two flash durations at eight different positions (in colors) compared to predicted model responses (gray) from an example cell (same as *Figure 1*). Model parameters were optimized to each cell using only responses from bars of width 2 (brown frame). (C) Peak measured response compared to the peak of the predicted response for all bar flash stimuli from all cells and positions (empty marker denotes short duration flash, filled denotes long flashes). Each dot represents the mean peak response (for top 10 of 1000 optimization solutions for each cell, estimated by fit error), while vertical lines (most obscured by markers) represent maximal and minimal values of the peak responses. Diagonal lines surrounding the unity line denote + /- the upper quartile of the Mean Absolute Deviation (MAD) of responses to repeated presentation of the same stimuli (see Materials and methods). (D) Mean measured responses to moving bars of three widths and two speeds overlaid with predicted responses from linear (superposition of flash responses) and EI conductance model for same example cell and the same model parameters as in (B). Note scale change on voltage axis of linear predictions due to overestimation. (E) Peak measured responses compared to peak predicted responses from both linear and EI models for all moving bars (grouped by bar width). Plotting conventions as in C; except PD in red, ND in blue. Note consistent overestimation of linear model, which is exacerbated for larger stimuli. See also *Figure 4—figure supplement 1* for distributions of individual cell measurement and EI model prediction responses.

The online version of this article includes the following figure supplement(s) for figure 4:

**Figure supplement 1.** Reliability of model predictions across cells.

stimuli and since these are sensitive to stimulus position within the RF, we optimized model parameters using responses from individual neurons, and not a population average.

The size of the stimuli used to probe directionally selective cells is a prominent confound when comparing results from different studies, which has resulted in conflicting interpretations about the possible mechanisms implementing DS. For example, a previous study using calcium imaging only observed PD enhancement in T5 cells for stimuli above 6° (*Haag et al., 2017*). Since we optimized our parameters using responses to width 2 bar flashes alone, we postulated that if increasing the width of the bar above a certain size evoked qualitatively different responses, we would see a clear discrepancy between predictions and recorded responses for width four bars, but not for width one bars. *Figure 4B* shows the results from an example cell, with the recorded responses and model prediction overlaid for three bar widths and with both fast and slow flash durations. The model predicts the magnitude and dynamics of the responses to all three widths, with no systematic difference for the larger or smaller width bars. The broad accuracy of these predictions can be seen when comparing the maximal response magnitude between model output and measurements across all the cells (*Figure 4C*). The accuracy of model predictions to measured data will, in part, be limited by trial-to-trial variability in the recorded responses to identical stimuli. We have used this variability as a simple bound on the accuracy of model predictions, by plotting the +/- upper quartile of the mean absolute deviation (MAD) across cells on either side of the diagonal (identity) line (*Figure 4C*). Since width two bars were used for optimizing model parameters, the spread around the diagonal is narrowest for this condition, with most simulation results falling within the 'MAD bounds' (see Materials and methods and *Figure 4—figure supplement 1*). However, the spread around the diagonal for width one or width four is only slightly broader, confirming that the responses across these bar widths can be predicted without further modifications to the model.

Our simple model accounts for responses to local flash stimuli of multiple widths (containing no directional information), but how well does it predict responses to moving bars of different widths? Our moving bar stimuli are comprised of an ordered sequence of bar flashes. This allowed us to compare EI model responses to moving bar stimuli with the linear superposition of bar flash responses (summed responses after appropriate temporal alignment). *Figure 4D* shows example traces from the same cell as in *Figure 4B*, with linear predictions on the left and EI model predictions on the right (same parameters as above). The linear model predictions capture some of the response dynamics but appear to overestimate response magnitudes, particularly for strong stimuli. Our EI model, on the other hand, captures both the magnitude of the maximal response in both directions and the dynamic structure of the response traces. For example, the simulated results to slow ND motion show a hyperpolarizing dip before the depolarizing peak, but this dip is absent from ND motion traces for fast motion. This is explained by the slow inhibitory conductance, which cannot contribute much to the early component of the response to fast motion. We again compared the maximal responses between both models' predictions and measured data across all cells. Predictions from the linear model consistently overestimate the measured response (*Figure 4E*, left), and this overestimation is most severe for fast responses to wide bars. Although the limitations of a linear model have already been shown in *Figure 3*, here we find an even larger discrepancy with predictions of moving bar response due to error accumulation over time. In contrast, the EI model predictions match the measured responses more closely, with comparisons of max responses amplitudes spread evenly around the diagonal (*Figure 4E*, right). Notably, if PD enhancement contributed to generating DS responses to moving bars of width four in our measurements, we would see a consistent underestimate of PD motion by the simulation. However, the EI predicted responses for both PD and ND motion show the same symmetrical spread around the diagonal (*Figure 4E*, right), again confirming that a single, simple non-linear mechanism can account for DS responses to moving bars of different widths.

## Conductance-based EI model recapitulates responses to more complex spatial and temporal stimuli

Given the strong correspondence between model predictions and response measurements for bar flashes and moving bars across speeds and widths, we challenged our EI model with more complex stimuli. Bar flashes have a simple spatial and temporal structure; moving bars are comprised of sequential activation of adjacent positions. The stimulus presented in the example in *Figure 5A*, flashes of gratings in different phases, has a non-contiguous spatial structure, requiring spatial

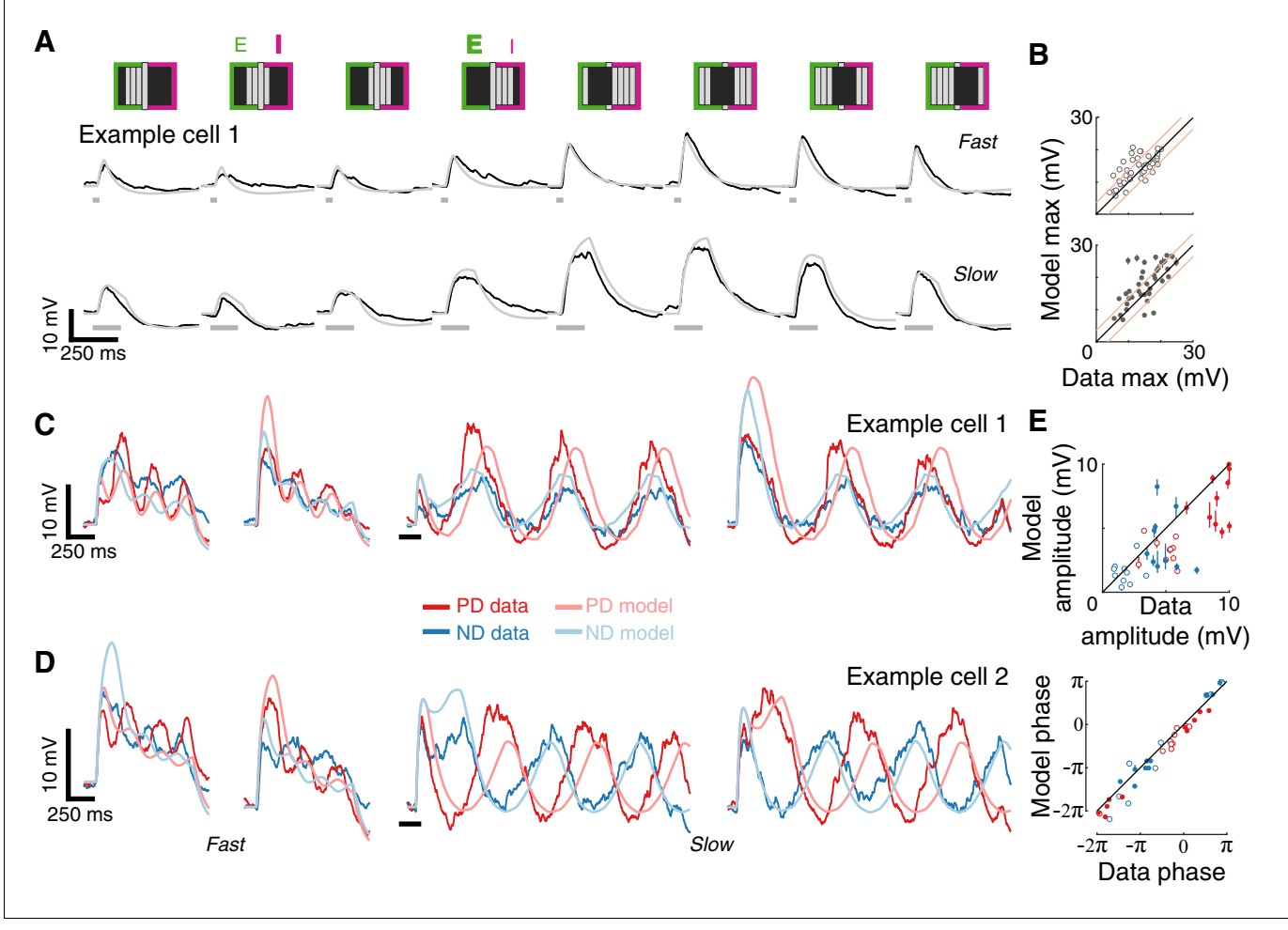

**Figure 5.** Conductance-based model recapitulates responses to more complex spatial and temporal stimuli. (A) Mean measured responses to fast (40 ms) and slow (160 ms) flashes of grating stimuli (dark and background brightness level) in different phases compared with model predictions (same example cell, same model parameters as in *Figure 4B*). Stimulus schematic above traces. In two phases, the expected relative contribution (bold >normal) of excitation and inhibition based on the position of the dark bars is denoted. (B) Peak measured response compared to peak predicted response, for flashes of all grating phases, grouped by flash duration. Plotting conventions as in *Figure 4C* (only results from cells with a cardinal PD-ND axis are shown; n = 5, see Materials and methods). (C) Mean measured responses to grating stimuli moving at two speeds (temporal frequency of 3.125 Hz and 0.78 Hz, 40 ms and 160 ms steps), with two different starting phases, compared to model prediction for the same example cell and model parameters as 4B. (D) As in (C), only from a different example cell. Note the difference in phase relations between PD and ND responses for cells in (C) and D). (E) Top: Fit amplitude values for measured response compared to the model predicted responses for all grating stimuli (see Materials and methods for details). Plotting conventions as in 4E. Bottom: fit phase values for measured responses compared to model prediction responses.

integration from different RF regions. As can be seen from the example cell (same cell and parameters as in *Figure 4*), our model also accurately predicted responses to these grating flashes. Importantly, when the visual input stimulated both the excitatory and the inhibitory fields (E or I above stimulus schematic in *Figure 5A*), the model predicted the dominant conductance, both in this individual example and in the population (*Figure 5B*).

Next, we challenged our model with drifting gratings, the most complex stimulus we presented. This stimulus has a non-contiguous spatial structure that is swept across the RF in a temporally cyclic manner. We simulated responses to square wave grating (composed of dark and background level bars) moving at two different speeds and starting from two different phases (*Figure 5C*). Although the model parameters were optimized using only the static stimuli responses (*Figure 4B*, middle), the model's predictions capture several aspects of the recorded responses: the transient responses to the appearance of the grating (that differ dramatically between the two phases), the amplitude of

the oscillation (compared in *Figure 5E*), and even the apparent amplitude adaptation in the responses to fast-drifting gratings. The model correctly predicted some surprising aspects of measured responses, such as the different phase relations between PD and ND responses in the different cells (most cells show out-of-phase responses like Example cell 2, *Figure 5D*, while a few showed in-phase responses, like Example cell 1, *Figure 5C*). The correspondence between prediction and measurement (*Figure 5E*) suggests that this dramatic phase difference is largely a consequence of the structure of the first-order RF of each cell. In summary, a simple model that integrates fast, excitatory and spatially offset, slow inhibitory inputs predicts T5 responses to a range of visual stimuli.

## A conductance model relying on removal of excitatory input cannot recapitulate T5 responses

In our previous paper (*Gruntman et al., 2018*), we applied the EI model to T4 neurons from the ON motion pathway. T4 neurons receive small-field, local inhibitory inputs at a dendritic location that corresponds to the trailing side in our EI model, but T5 neurons do not (*Shinomiya et al., 2019*; *Davis et al., 2018*). Therefore, we asked whether T5 responses can be modeled without local inhibitory inputs, but rather with the local removal of excitation. The parameters for this model (referred to as the $E_+E_-$ model) include the same linear filters that set the temporal dynamics of the conductances and the same Gaussian filters that establish their spatial receptive fields. To instantiate a removal of excitation model, we included two additional parameters: a magnitude for a stimulus-independent excitatory conductance from which excitation can be removed, and a magnitude for a stimulus-independent inhibitory conductance that ensures the neuron's baseline will not drift due to the addition of the above conductance (see Materials and methods and *Supplementary file 2*).

*Figure 6* shows $E_+E_-$ model simulation results (same optimization procedure) for the same example cell as in *Figure 4*. Although the $E_+E_-$ model predicts the response magnitude for the bar width two stimuli that were used in the optimization, it fails to generalize, as evidenced by poor estimates of responses to wider bar flash stimuli (*Figure 6A,B*). The $E_+E_-$ model predictions for moving bar stimuli also consistently overestimate the measured responses, with larger errors in the responses to ND stimuli (*Figure 6C*). The differential overestimation of ND responses means that response magnitudes to bars moving in the preferred or the null direction are similar. As a consequence, the directional selectivity of the $E_+E_-$ model is lower than measured (DSI for the example cell indicated in *Figure 6C*).

In analyzing the low directional selectivity of the $E_+E_-$ model, we found that it is, in part, due to saturation of the removed excitatory conductance. To expand the effective dynamic range of the $E_-$ conductance we tried several alternative optimization procedures (*Figure 6—figure supplements 1* and *2*). We were able to reproduce responses to the stimuli used for optimization, but the same parameters failed to generalize to other stimuli. Specifically, the $E_+E_-$ models generated responses that were less directionally selective than both our measurements and the EI model.

## Inhibition is superior to removal of excitation for generating directional selectivity

To understand the inability of the $E_+E_-$ model to generate strong directional selectivity for moving bars, we compared it to our EI model under similar conditions. We found three major differences contributing to the poor performance of the $E_+E_-$ model (*Figure 7*, the relevant comparison for each major difference is indicated with a circled number).

1. *The removal of excitation is bounded.* Both models rely on a balance between a depolarizing conductance, which is excitatory in both, and a hyperpolarizing conductance, which is implemented with inhibition in EI model and the removal of excitation in $E_+E_-$ model. Both optimized model solutions reduce the (temporal) overlap between the depolarizing and hyperpolarizing conductances for a bar moving in the preferred direction, and maximize the overlap when the bar moves in the null direction (*Figure 7*, PD and ND). Whereas the depolarizing and hyperpolarizing conductances in the EI model are of similar magnitude, the hyperpolarizing conductance in the $E_+E_-$ model is only half the magnitude of the depolarizing conductance. This difference stems from an intrinsic limitation of using the removal of an excitatory conductance as the mechanism for hyperpolarization. An excitatory conductance can only be removed from an activated (open) conductance (*Figure 7*, middle column, indicated by the dashed line). Furthermore, $E_-$ cannot exceed the magnitude of the activated

conductance, resulting in an upper bound on $E_-$ (see Materials and methods for details). Since this limit on the removed conductance is intrinsic to the model, it cannot be overcome with different optimization procedures (*Figure 6—figure supplements 1* and *2*).

2. *Total normalizing conductance decreases with increasing hyperpolarization.* Is the $E_-$ bound the primary difference between the models, or do other factors also contribute? We analyzed a 'Scaled EI' model that was constructed to match the depolarizing and countering hyperpolarizing conductances of the $E_+E_-$ model, a manipulation that eliminated the differences stemming from the $E_-$ bound. While both models exhibit similar conductance changes, the Scaled EI model produces better directional selectivity than the $E_+E_-$ model (seen in the voltage traces, *Figure 7*, bottom), suggesting additional contributions to the EI model's superior performance. In both models, the voltage responses depend on the weighted difference between the depolarizing and hyperpolarizing conductances (integrated conductance, $\Delta G$), normalized by the total conductance ($\Sigma G$). By construction, the value of the integrated conductance is identical in the Scaled EI and $E_+E_-$ models (*Figure 7*, $\Delta G$ row). However, the normalizing conductance is very different. In the Scaled (and original) EI model, hyperpolarization is achieved by adding an inhibitory conductance, making the total conductance the sum of the depolarizing and large hyperpolarizing conductances ($\Sigma G = 1+E+I$). In the $E_+E_-$ model hyperpolarization is achieved by removal of excitation, making the total conductance the difference between the two ($\Sigma G = 1+ E_+- E_-$). This effect is evident in the smaller values of $\Sigma G$ of the $E_+E_-$ model (*Figure 7*, $\Sigma G$ row).

3. *Non-linearity in the $E_+E_-$ model cannot differentially affect PD and ND.* As noted in the previous point, in the $E_+E_-$ model the hyperpolarizing conductance identically contributes to both the integrated ($\Delta G$) and the total conductance ($\Sigma G$). Consequently, a peak in the time course of the integrated conductance is always normalized to a corresponding peak of the total conductance (a 1 degree of freedom non-linearity), regardless of the direction of input motion. By contrast, in both EI models, the difference in $\Delta G$ is normalized with the sum in $\Sigma G$. This distinction has a profound effect on directional selectivity, since the hyperpolarizing conductance is less coincident with the depolarizing conductance in response to PD stimuli (*Figure 7*, PD) compared to ND stimuli (*Figure 7*, ND). This increased overlap between the conductances for ND stimuli leads to increased suppression in the voltage response (*Figure 7*, vertical lines connecting $\Delta G$ peak to $\Sigma G$ curve).

This model comparison shows that removing excitation is qualitatively different from adding inhibition. Implementing this seemingly simple change from I to $E_-$ leads to changes in the conductance balance, the creation of an intrinsic bound on the strength of hyperpolarization, and a less efficient differential suppression of PD and ND responses. This analysis further clarifies the importance of non-linearity for computing directional selectivity in our EI model. By normalizing the difference between the depolarizing and hyperpolarizing conductances with their sum, the EI model amplifies the differences between the PD and ND responses (already present in the $\Delta G$). Taken together, these modeling results corroborate our central finding that inhibition appears to sculpt the essential asymmetry for directionally selective responses in T5 cells.

## Discussion

In this study we used whole-cell recordings of T5 cells, the OFF directionally selective neurons in *Drosophila* (*Figure 1*) to uncover the mechanism underlying the generation of directionally selective motion responses. Using local bar flashes, we mapped the first-order receptive field of T5 neurons and revealed an asymmetric spatial structure, consisting of offset depolarizing and hyperpolarizing input fields (*Figure 2*)—the very same structure we previously described for T4 (ON directionally selective) neurons (*Gruntman et al., 2018*). Using pairs of bar flashes, we mapped the responses to second-order stimuli, and found no amplifying pairwise interactions that are indicative of a PD enhancing mechanism, rather we only found evidence for ND suppression (*Figure 3*). We employed a slightly simplified version of our previous biophysical model (*Gruntman et al., 2018*), now applied to single-neuron data, to accurately predict T5 responses to static and moving bars of different widths, and even static and drifting gratings (*Figures 4* and *5*). Since T5 neurons are not known to have small-field inhibitory inputs, we explored a related computational model based on the removal of excitation rather than direct inhibition, and found that this alternative poorly reproduces DS responses and is an inadequate model of T5 neurons (*Figures 6* and *7*).

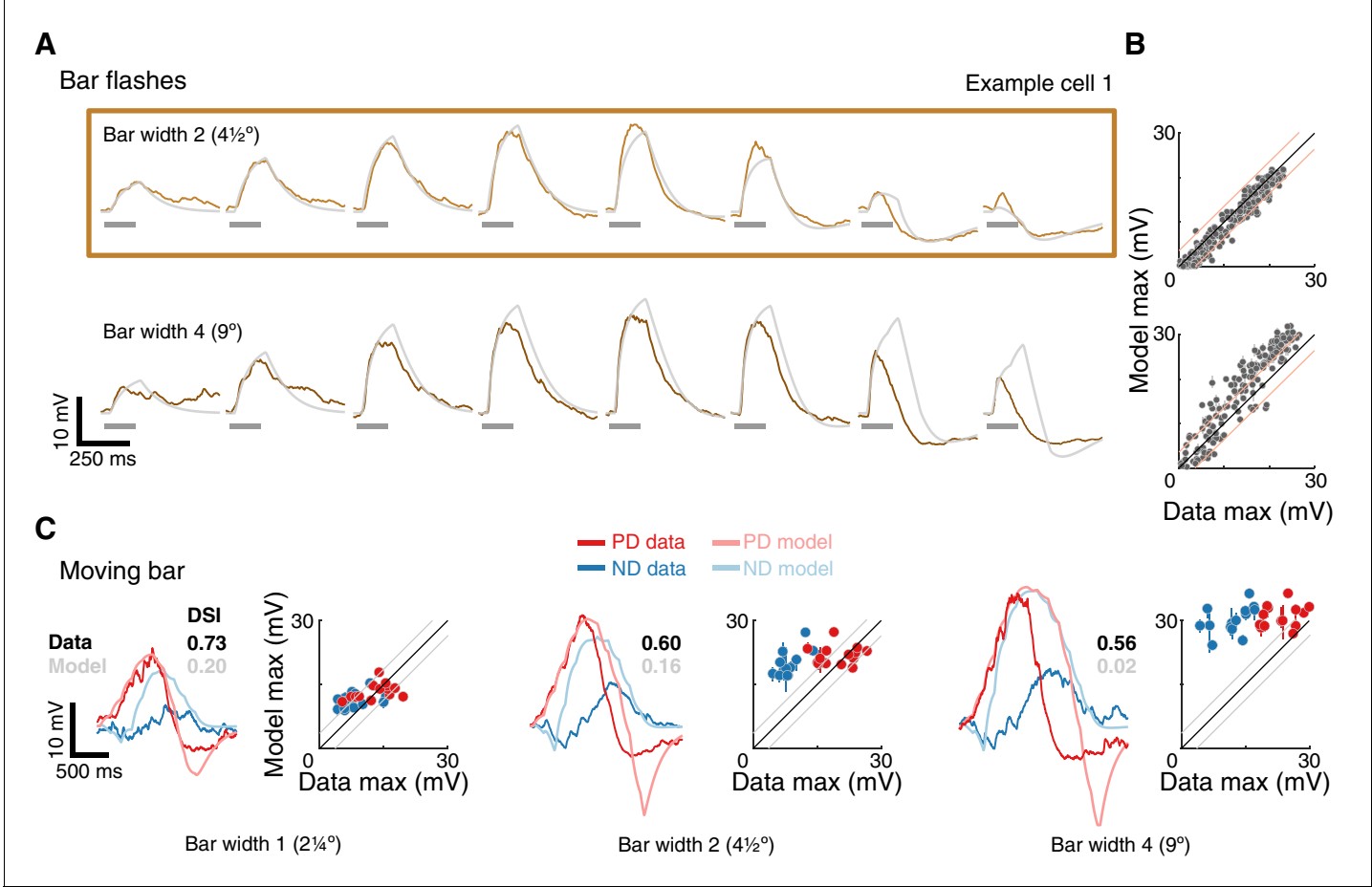

**Figure 6.** A conductance model relying on removal of excitatory input cannot recapitulate T5 responses. (**A**) Mean measured responses to single bar flashes of two widths flashed for 160 ms at eight different positions from the same example cell as in *Figure 4* (in colors) compared to predicted E$_+$E$_-$ model responses (gray). Model parameters were optimized using only responses to bar width two flashes (brown frame). (**B**) Peak measured response compared to peak predicted response for bar flash stimuli from all cells and positions. Same conventions as in *Figure 4*, but only 160 ms flash responses are shown. The E$_+$E$_-$ model overestimates the width 4 bar flash responses. (**C**) Traces: Mean measured responses from same example cell as in *Figure 4* for slow moving bar stimuli overlaid with E$_+$E$_-$ model predicted responses. Scatter plots: Peak measured responses compared to peak predicted responses from E$_+$E$_-$ model for all moving bar responses across cells (grouped by bar width). Plotting conventions as in *Figure 4*. Note similar response magnitude for PD and ND stimuli; DSI comparison provided for the example cell and E$_+$E$_-$ model. See also *Figure 6—figure supplements 1* and *2* for E$_+$E$_-$ model results from different optimization procedures.

The online version of this article includes the following figure supplement(s) for figure 6:

**Figure supplement 1.** E$_+$E$_-$ model relying on removal of excitatory input cannot recapitulate T5 responses, even when optimized with bars of width 2 and 4.

**Figure supplement 2.** E$_+$E$_-$ model relying on removal of excitatory input cannot recapitulate T5 responses, even when optimized with flashing and moving bars.

The first-order RF of T5 neurons, with a broad depolarization field leading a narrow trailing hyperpolarization field, strongly resembles the structure we uncovered for T4 neurons (*Figure 2D*). This RF characterization agrees with recent measurements made using voltage imaging of T5 neurons (*Wienecke et al., 2018*), but augments them with absolute response magnitudes and time constants. The second-order response mapping was used as an explicit test for non-linear interactions that facilitate motion detection (*Figure 3*). Across a very large set of stimuli, we find a clear pattern in the responses: no evidence for an amplifying non-linearity, but abundant evidence for sublinear integration on the trailing side of the RF due to asymmetric hyperpolarizing inputs. A recent study reported PD enhancement only occurs for larger (>6°) stimuli (*Haag et al., 2017*), however, even with wide bar stimuli (that nearly cover the T5 RF), we did not observe any instances of PD

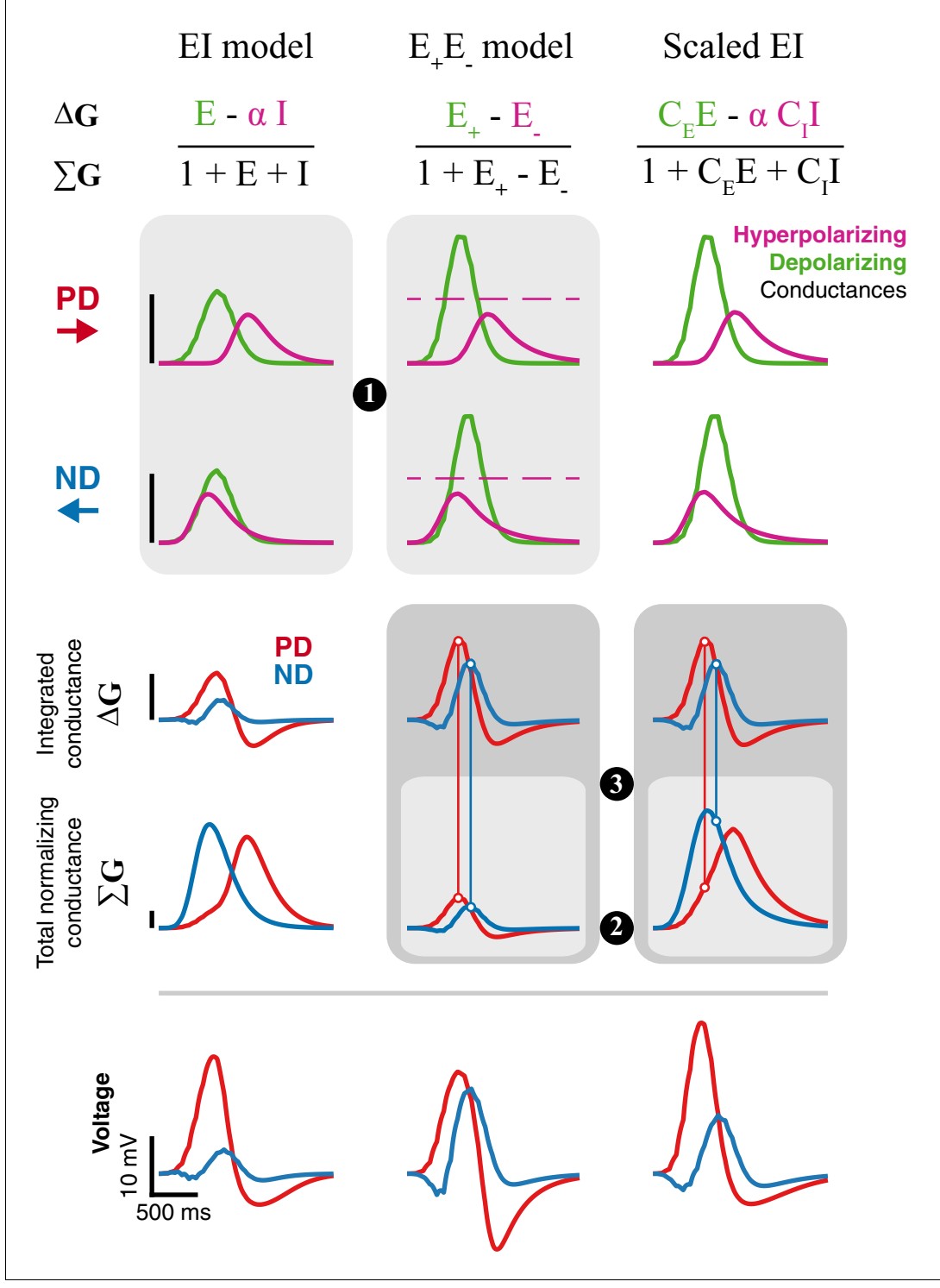

**Figure 7.** Inhibition is superior to removal of excitation for generating directional selectivity. Each column presents responses from a different model version to a moving bar stimulus of width 2 with 80ms step duration: EI model from *Figure 4*, E$_+$E$_-$ model from *Figure 6*, and a Scaled version of the EI model. Parameters for first and second column are the best fit (for each model, separately) for the example cell used in *Figure 4* and *6*. The Scaled EI model parameters were inherited from the E$_+$E$_-$ model (see Materials and methods for details), with coefficients (C$_E$, C$_I$) required to match the depolarizing and hyperpolarizing conductances in the numerator of the E$_+$E$_-$ model. First and second rows show the dynamics of the depolarizing and hyperpolarizing conductances in each model to PD and ND stimuli. The dashed line in the E$_+$E$_-$ column represents the magnitude of $g_E^*$ (maximal value for E$_-$, see

*Figure 7 continued*

Materials and methods). Third, fourth, and fifth rows show, respectively, the integrated conductance, the total conductance, and the voltage response of each model version. Circled numbers and shaded frames emphasize comparisons discussed in the Results. Scale bars are shared across columns and, for the top four rows, represent 0.5 normalizing conductance (Leak for EI and scaled EI, total stimulus-independent conductances for $E_+E_-$).

enhancement in the membrane potential responses (*Figure 3D,E* and *Figure 3—figure supplements 1* and *2*).

While the computational mechanism appears to be similar between T4 and T5 neurons, we did measure some differences between these cell types. T5 responses showed a position-dependent rise time that was slower on the leading side than on the trailing side (*Figure 2C*), whereas T4 neurons did not (Figure 2D of *Gruntman et al., 2018*). Since T5 dendrites are retinotopically aligned with the preferred direction of motion, and since different input types synapse onto different locations of the T5 dendrite (*Shinomiya et al., 2019*), it is possible to connect the available functional data from these input types (*Arenz et al., 2017*; *Serbe et al., 2016*) to their input position. Indeed, Tm9 neurons, that synapse onto the distal branches of T5 dendrites (corresponding to the leading side of the RF), are also the slowest among the four major input types (*Arenz et al., 2017*). However, we found no evidence that this spatial change in rise time plays any functional role in the computation of DS. The depolarizing part of the RF is 'motion blind' (*Figure 3*) and when we modified our model to account for this temporal difference, we recorded no substantial improvement in model performance (results not shown). Perhaps a more important difference between T4 and T5 neurons is the sensitivity to weak stimuli. Responses to single bar flashes in T4 were roughly equivalent to T5 responses to stimuli of twice the width (*Figure 2D*). *Haag et al. (2017)* also reported a similar difference. Since this difference is already measured in response to non-moving bar flashes, it is likely a difference in the response sensitivity of upstream neurons. This difference may also suggest different detection thresholds in the ON and OFF motion pathways, which would in turn suggest different behavioral responses to small moving dark and bright objects.

In our previous T4 work, we implemented a simple computational model that integrates spatially offset excitatory and inhibitory conductances in a passive, biophysical model (*Gruntman et al., 2018*). In this study we used a slightly simplified version of this EI model to predict single neuron responses to a range of stimuli (*Figures 4* and *5*). Importantly, we showed that a model optimized only to predict single object impulse responses also accurately predicts moving bar and drifting grating responses (*Figures 4* and *5*). This EI model concretely instantiates an algorithmic mechanism while also suggesting a biological implementation for the ND suppressing nonlinearity.

T4 and T5 share a high degree of anatomical similarity (*Fischbach and Dittrich, 1989*; *Shinomiya et al., 2019*; *Strausfeld and Lee, 1991*; *Takemura et al., 2017*; *Takemura et al., 2013*). Focusing only on their dendrites, both neuron types receive four major columnar inputs, both preferentially synapse onto neurons of their same subtype, and both receive prominent input from CT1, a single, large GABAergic interneuron (*Shinomiya et al., 2019*). However, the described anatomy of the T4 and T5 inputs, combined with transcriptomic data (*Davis et al., 2018*), shows that T4 receives mixed inputs from columnar cholinergic, GABAergic, and Glutamatergic cells, while all columnar inputs to T5 are cholinergic. This substantial difference between the inputs of these two cells is especially surprising since the DS generation mechanism in T4 and T5 appears to be all but identical: fast depolarization with asymmetric slow hyperpolarization on the trailing side of the RF.

In light of this anatomical difference, we explored a related model class, where instead of direct inhibition, the model neuron integrates an excitatory input together with an offset excitatory input which is reduced from a tonic baseline level in a stimulus-dependent manner. While this $E_+E_-$ model is able to capture several aspects of the first-order receptive field, it does not generalize well, and is an inferior predictor of moving bar responses (*Figure 6* and *Figure 6—figure supplements 1* and *2*). By analyzing the responses of the EI and $E_+E_-$ model, we identified multiple factors that explains why direct integration of excitatory and inhibitory inputs results in superior directional selectivity (*Figure 7*). Our analysis suggests that T5 neurons are very likely to receive inhibitory inputs that respond to local visual stimuli. A potential source for this input is CT1 that was thought to provide only non-local inhibitory inputs due to its morphology. However, a recent study raises the possibility that CT1 may function like a small-field, columnar neuron and provide the necessary local inhibitory

input (*Meier and Borst, 2019*). In agreement with this proposal, transcriptomic profiling shows that T5 neurons express several GABA receptors (*Davis et al., 2018*). A further possibility is that some fraction of the cholinergic inputs could activate an inhibitory conductance (first proposed in *Shinomiya et al., 2014* ), in which case the T4 model should apply to T5, but be understood to make use of a rather different molecular implementation. While elaborations of our T5 model could be pursued to gain further insight on potential implementations, we believe that more experimental evidence is required to establish the source and role of inhibition in the T5 computation.

In the current study, we expanded the set of visual stimuli we delivered, which allowed us to reconcile our results with those of other groups. First, we found no evidence for a PD-enhancing mechanism in T5 neurons, even with stronger stimuli as was previously shown with calcium imaging (*Haag et al., 2017*). A potential resolution to this discrepancy may be the differences between voltage and calcium responses in the same neurons. When both calcium and voltage responses were

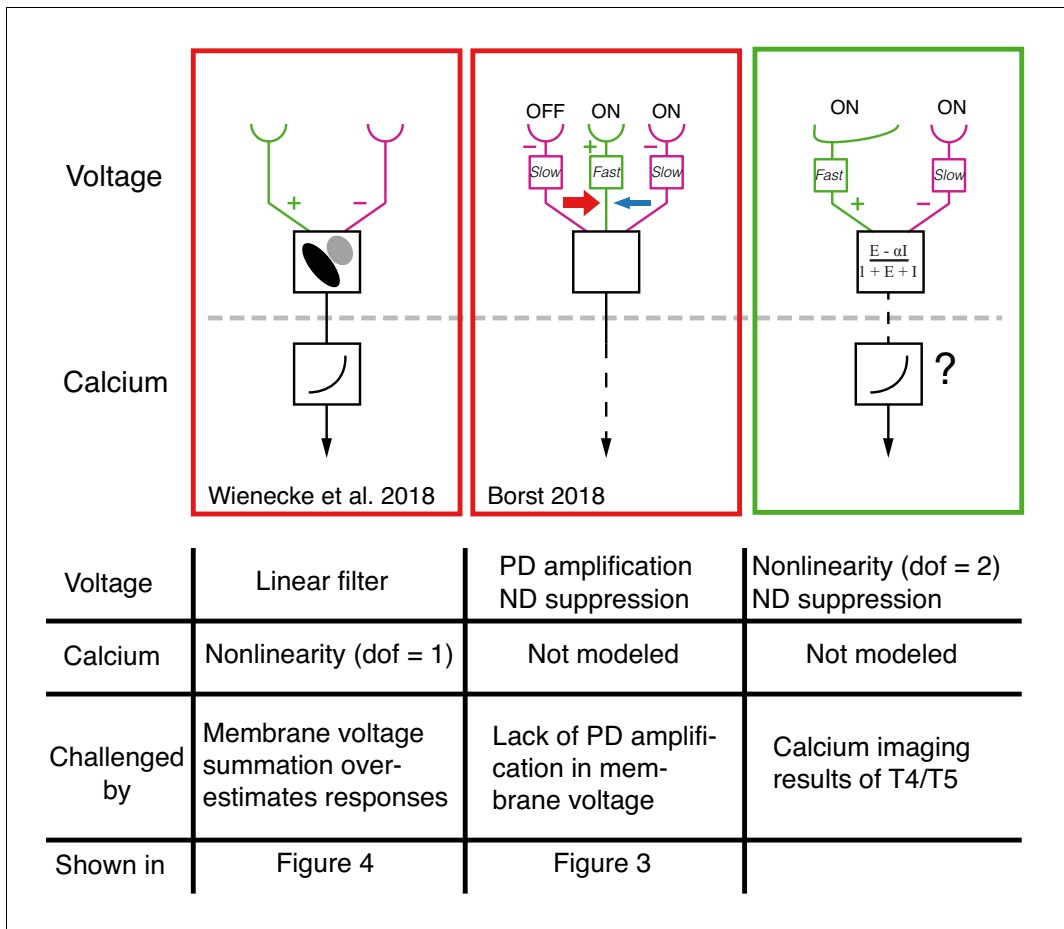

| | | | |
|---|---|---|---|
| Voltage | Linear filter | PD amplification ND suppression | Nonlinearity (dof = 2) ND suppression |
| Calcium | Nonlinearity (dof = 1) | Not modeled | Not modeled |
| Challenged by | Membrane voltage summation over-estimates responses | Lack of PD amplification in membrane voltage | Calcium imaging results of T4/T5 |
| Shown in | Figure 4 | Figure 3 | |

**Figure 8.** Comparison between current models for T4/T5 computation of directional selectivity. One recent model class, represented here by the proposal of *Wienecke et al. (2018)* uses a tilted linear spatio-temporal filter to represent voltage responses, which are then followed by a nonlinear voltage-to-calcium transformation (1 degree of freedom, dof). Another current model class, represented here by the version from *Borst (2018)*, proposes a three-arm model that uses a biophysical mechanism to generate PD amplification and ND suppression; the calcium transformation is not modeled. Both of these models cannot explain our T5 results. The specific results that challenge these proposals are detailed in the table below each model. The right column shows a unifying proposal in which our EI model implements a 2 dof nonlinearity (dofs represented by $\Delta G$ and $\Sigma G$, see *Figure 7*) that results in ND suppression in the voltage signal. This nonlinearity, which is a consequence of the biophysical model that captures T5 responses, enables greater selectivity than a single dof transformation (for intuition, see *Figure 7*). Since our model does not explain the PD enhancement observed in calcium measurements (e.g. *Fisher et al., 2015*; *Salazar-Gatzimas et al., 2016*; *Haag et al., 2016*), we augment this model with a voltage-to-calcium transformation.

imaged in T5 neurons, *Wienecke et al. (2018)* found amplification only in the calcium signal, and this amplification was proposed to emerge from the transformation of membrane voltage to fluorescent calcium signals. However, the model proposed by *Wienecke et al. (2018)* is based on a voltage signal that is integrated linearly across the cell's receptive field; this linear signal is then amplified or suppressed by a 1 degree of freedom, non-linear, voltage-to-calcium transformation (*Figure 8*, left column). Our electrophysiological measurements show that the voltage signals do not simply integrate linearly but are reliably sublinear (see for example *Figures 3* and *4*).

Another form of current directional selectivity models, represented by several recent studies (*Borst, 2018*; *Haag et al., 2016*; *Strother et al., 2017*) have proposed a 3-arm detector that generates ND suppression by integrating the middle and trailing arms and PD enhancement by integrating the leading and middle arms. Although the evidence for this form of PD enhancement comes from calcium imaging, the model in *Borst (2018)* uses a biophysical framework to generate PD enhancement in the voltage signal (*Figure 8*, middle column). Our data shows no evidence for this form of PD enhancement (*Figure 3D,E* and *Figure 3—figure supplement 1*). Instead, we propose the following unifying framework (*Figure 8*, right column): the voltage response of T5 (and T4) neurons to OFF (and ON) stimuli is generated by offset excitatory and inhibitory inputs. This integration has the effect of introducing a nonlinearity (with 2 degrees of freedom) that is a potent and efficient mechanism for generating directional selectivity. Subsequent to this, a secondary mechanism that relates the transformation of membrane voltage to calcium in the axon terminals could further enhance signals for PD motion and suppress signals for ND motion using an additional nonlinearity.

Motion vision in flies has long been a model system for neuronal computation. It is implemented by a small-scale circuit that is genetically accessible, is a simple yet foundational computation for vision, and has a well-defined behavioral role. Potential algorithms for this computation were already suggested in the middle of the last century. These algorithms were designed to explain behavioral responses and were refined to match responses of neurons downstream of T4 and T5. However, this approach was under constrained because the available measurements were not from the neurons implementing directional selectivity. We have proposed a biophysical model for T4 and T5 neurons that is constrained by high-resolution electrophysiological measurements from these very same neurons. By minimizing the slack between model implementation and experimental measurements, we have ruled out many alternative models and proposed a parsimonious, robust, and broadly predictive algorithm for motion vision computation.

# Materials and methods

## Key resources table

| Reagent type (species) or resource | Designation | Source or reference | Identifiers | Additional information |
|---|---|---|---|---|
| Genetic reagent (*D. melanogaster*) | UAS-GFP | Janelia Research Campus | pJFRC28-10XUAS-IVS-GFP-p10 (attP2) | Rubin Lab JFRC28 |
| Genetic reagent (*D. melanogaster*) | Stable split Gal4 (T5) | Janelia Research Campus | w; VT055812-AD(attP40); R47H05-DBD(attP2) | Rubin Lab SS25175 |
| Software, algorithm | MATLAB | Mathworks Inc | 2018b | |

## Electrophysiology

Experiments were performed on 1–2 day old female *Drosophila melanogaster* (flies were reared under 16:8 light:dark cycle at 24°C). To target T5 cells, a single genotype was used: pJFRC28-10XUAS-IVS-GFP-p10 (*Pfeiffer et al., 2012*) in attP2 crossed to stable split-GAL4 SS25175 (w; VT055812-AD(attP40); R47H05-DBD(attP2)) generously provided by Aljoscha Nern in Gerry Rubin's lab (line details with expression data available from http://splitgal4.janelia.org/). Flies were briefly anesthetized on ice and transferred to a chilled vacuum holder where they were mounted, with the head tilted down, to a customized platform machined from PEEK using UV-cured glue (Loctite

3972). CAD files for the platform and vacuum holder are available upon request. To reduce brain motion the proboscis was fixed to the head with a small amount of the same glue. The posterior part of the cuticle was removed using syringe needles and fine forceps. The perineural sheath was peeled using fine forceps and, if needed, further removed with a suction pipette under the microscope. To further reduce brain motion, muscle 16 (*Demerec, 1950*) was removed from between the antenna.

The brain was continuously perfused with an extracellular saline containing (in mM): 103 NaCl, 3 KCl, 1.5 CaCl$_2$ 2H$_2$O, 4 MgCl$_2$ 6H$_2$O, 1 NaH$_2$PO$_4$ H$_2$O, 26 NaHCO$_3$, 5 N-Tris (hydroxymethyl) methyl-2- aminoethane-sulfonic acid, 10 Glucose, and 10 Trehalose (*Wilson and Laurent, 2005*). Osmolarity was adjusted to 275 mOsm, and saline was bubbled with 95% O$_2$/5% CO$_2$ during the experiment to reach a final pH of 7.3. Pressure-polished patch-clamp electrodes were pulled for a resistance of 9.5–10.5 MΩ and filled with an intracellular saline containing (in mM): 140 KAsp, 10 HEPES, 1.1 EGTA, 0.1 CaCl2, 4 MgATP, 0.5 NaGTP, and 5 Glutathione (*Wilson and Laurent, 2005*). 250 µM Alexa 594 Hydrazide was added to the intracellular saline prior to each experiment, to reach a final osmolarity of 265 mOsm, with a pH of 7.3.

The mounted, dissected flies were positioned on a rigid platform mounted on an air table. Recordings were obtained from labeled T5 cell bodies under visual control using a Sutter SOM microscope with a 60X water-immersion objective. To visualize the GFP labeled cells, a monochrome, IR-sensitive CCD camera (ThorLabs 1500M-GE) was mounted to the microscope, an 850 nm LED provided oblique illumination (ThorLabs M850F2), and a 460 nm LED provided GFP excitation (Sutter TLED source). Images were acquired using Micro-Manager (*Edelstein et al., 2014*), to allow for automatic contrast adjustment.

All recordings were obtained from the left side of the brain. Current clamp recordings were sampled at 20 KHz and low-pass filtered at 10 KHz using Axon multiClamp 700B amplifier (National Instrument PCIe-7842R LX50 Multifunction RIO board) using custom LabView (2013 v.13.0.1f2; National Instruments) and MATLAB (Mathworks, Inc) software. Shortly after breaking in, recordings were stabilized with a small injection of a hyperpolarizing current (0–3 pA) setting the membrane potential to a range between −60 to −55 mV (uncorrected for liquid junction potential). Occasionally, the injected current required adjustments, but these were done prior to the acquisition of the single bar flash data. To verify recording quality, current step injections were performed at the beginning of the experiment.

## Visual stimuli

The display was constructed from an updated version of the LED panels previously described (*Reiser and Dickinson, 2008*). The arena covered slightly more than one half of a cylinder (216° in azimuth and ~72° in elevation) of the fly's visual field, with the diameter of each pixel subtending an angle of (at most) 2.25° on the fly eye. Green LEDs (emission peak: 565 nm) were used, dark stimuli (off pixels) were presented on an intermediate intensity background of ~31 cd/m$^2$.

Visual stimuli were generated using custom written MATLAB code that allowed rapid generation of stimuli based on individual cell responses. In contrast to the published stimulus control system (*Reiser and Dickinson, 2008*), we have now implemented an FPGA-based panel display controller, using the same PCIe card (National Instrument PCIe-7842R LX50 Multifunction RIO board) that also acquired the electrophysiology data. This new control system (implemented in LabView) streams pattern data directly from PC file storage, allowing for on-line stimulus generation. Furthermore, this new control system featured high precision (10 µs) timing and logging of all events, enabling reliable alignment of electrophysiology data with visual stimuli.

To map the receptive field (RF) center of each recorded cell, three grids of flashing dark squares (on the same intermediate intensity background) were presented at increasing resolution. Each flash stimulus was presented for 200 ms. First, a 6 × 7 grid of non-overlapping 5 × 5 LEDs (~11°×~11°) dark squares was presented (*Figure 1C*). If a response was detected, a denser 3 × 3 grid with 50%-overlapping 5 × 5 LEDs (~11°×~11°) bright and dark squares (to further verify these were T5 Cells) was presented at the estimated position of the RF center. If a recorded cell was consistently responsive to the first two mapping stimuli, a third protocol was presented to identify the RF center. A 5 × 5 grid of 3 × 3 LED bright squares separated by one pixel-shifts was presented at the estimated center of the second grid stimulus. The location of the peak response to this stimulus was used as the RF center in subsequent experiments. Once the RF center was identified, the moving bar

stimulus was presented in 8 directions with 80 ms step duration (equivalent to ~28°/s). The bar was nine pixels in height and 1, 2, or 4 pixels in width (results in *Figure 1D,E*). When moving in the cardinal directions, the motion spanned nine pixels. In the diagonal directions bar motion included more steps to cover the same distance (9 steps vs. 13 steps; see also *Figure 2—figure supplement 1*). Once the preferred direction had been estimated, dark bar flashes were presented on the relevant axis for widths 1,2 and 4. To verify full coverage of RF, this stimulus was presented over an area larger than the original motion window (at least 13 positions; results in *Figure 2*). In addition to these stimuli, most cells were also presented with additional stimuli following this procedure. All stimuli were presented in a pseudorandom order within stimulus blocks. All stimuli were presented three times, except for single bar flashes which were repeated five times. The inter-stimulus interval was 500 ms for moving stimuli and 800 ms for single bar flashes (to minimize the effect of ongoing hyperpolarization on the responses to subsequent stimuli).

Other presented stimuli were:

1. *Moving bar*. after identifying the PD-ND axis, moving bar stimuli were presented along this axis using either 40 ms or 160 ms steps (equivalent to 56°/sec or 14°/sec respectively). Bar height was the same for the mapping stimuli and width was either 1,2 or 4 pixels (corresponding to 2.25, 4.5 or 9°). Results in *Figure 4*.
2. *Apparent motion*. Bar pairs were presented in two different configurations. Either bars were of width two and the delay between the first and the second bar was adjusted to maintain fixed speed (i.e. correcting the temporal delay to account for the spatial difference in positions), or bars were of width four and the second bar was presented directly after the first, regardless of positional difference. This second configuration was meant to elicit the strongest responses possible. Results in *Figure 3* and *Figure 3—figure supplements 1* and *2*.
3. *Flashes of different grating phases*. Square wave gratings were of a constant spatial frequency (4 pixels OFF/4 pixels background intensity), presented in the same window size as the moving bar stimulus (nine steps for cells with PD along a cardinal direction, 13 steps for cells with PD along the diagonals). 8 phases of the grating stimuli were presented for 40 ms or 160 ms. Results in *Figure 5*.
4. *Moving grating*. Square wave grating with the same properties as above were presented with phases moving either in the forward or the reverse direction (PD or ND), and the initial phase presented being either 0 or 0.5. Results in *Figure 5*.

## Analysis

All data analysis was performed in MATLAB using custom written code. Since the T5 baseline was typically stable, we included only trials in which the mean pre-stimulus baseline did not differ from the overall pre-stimulus mean for that group of stimuli by more than 10 mV. We also verified that the pre-stimulus mean and overall mean for that trial did not differ by more than 15 mV (or 25 mV for slow moving bars, due to their strong responses). This is the same criteria we used for our previous T4 study (*Gruntman et al., 2018*). Responses were later aligned to the appearance of the bar stimulus and averaged (or the appearance of the bar in the central position in case of the 8-orientation moving bar). T5 cells are expected to signal using graded synapses. Consistent with this expectation, we find that T5 recordings only occasionally feature very small, fast transients (~1–2 mV in size) that could not be verified as spikes. Therefore, we have focused our analysis on the graded (sub-threshold) components of T5's responses.

Determining PD: After presenting the cell with 1 and 2 pixel wide bars moving in eight different directions at 80 ms per step (speed which was optimal for determining directional selectivity for T4 cells; *Gruntman et al., 2018*), the preferred direction for the cell was determined by a visual estimate of the responses to determine the middle of the relatively wide range of large responding directions (see *Figure 1*). Because stimuli were presented in 45° intervals, and the tuning of T5 neurons to direction is relatively wide, the more precise method for PD estimation that was used for T4 cells (*Gruntman et al., 2018*) was unnecessary.

DSI calculation: direction selectivity index was defined as DSI = $[PD_{max} – ND_{max}]/PD_{max}$, with each response max defined as the 0.995 quantile (a more robust estimate of the max than peak) within the stimulus presentation window.

*Single Position Flash Response – depolarization*: responses were defined as the 0.995 quantile (a robust estimate of the max) of the response during the time between bar appearance and flash

duration + 75 ms. If this number did not exceed 2.5 standard deviations of the pre-stimulus baseline, the response was defined as zero. For bars of width 2 and 4 the threshold was 2.7 and 2.9 standard deviations, since the responses were stronger. Standard deviation of the baseline was determined by fitting a gaussian to the pre-stimulus baseline values for all the stimulus presentations and extracting the sigma value from the fit.

*Single Position Flash Response - hyperpolarization*: same as above only the time window used was until the end of the trial (due to slower time course for hyperpolarization) and lower thresholds were used (1.5, 1.7, and 1.9 standard deviations, due to lower magnitude of hyperpolarization). These calculations were used for *Figure 2C,D*.

*Rise start/Rise time Calculation*: Only presentations in which the average SPFR was detected as depolarizing were used for this calculation. Start time was defined as the time from stimulus presentation to 10% of the of the value of the maximal response for that position. Rise time was defined as the time from 10% of the response maximum to 50% of the maximum. The data in *Figure 2C* are plotted as relative to the center (0) position, since the above values for the center position have been subtracted from all other positions for each cell separately. For each position in *Figure 2C* at least 10 cells passed the selection criteria.

*Apparent motion linear approximation (superposition)*: Single bar flash responses were aligned to the time of the corresponding position appearance in the apparent motion stimulus (bar pairs in *Figure 3* and moving bars in *Figure 4*). Responses were padded with zeros (since all were baselines subtracted) to extend brief single bar responses to the timescale of apparent motion. This procedure was used for the linear estimation throughout *Figure 3*, and in *Figure 4D*.

*Mean absolute deviation (MAD) calculation*: For each stimulus presentation the mean absolute deviation was calculated between the maximum of the mean response to the stimulus, and the maximum of the individual repeats. Once a maximal response was identified for the mean response, the maximal response was found for each corresponding repeat within a 100 ms window surrounding the mean peak response. We verified that there was no relationship between the maximal response magnitude and the MAD magnitude (by calculating MAD for different magnitude range) and pooled all the MAD estimates from all the different stimuli and all the cells for a global estimate. *Figure 4C, E* and *Figure 5B,E* are using the upper quartile for a MAD estimate for all the cells. In *Figure 4—figure supplement 1* we use the upper quartile from all the stimuli for each neuron separately.

*Data selection*: For *Figure 5*, only cells that showed a non-diagonal preferred direction (n = 5) are presented. Since we regard the receptive field as one dimensional (PD-ND axis) and since our LED arena is limited when generating diagonal stimuli, we chose to focus only on non-diagonal (cardinal) PD cells instead of fitting a second dimension to the receptive field of diagonal cells.

*Phase and amplitude calculation*: For *Figure 5*, we calculated the phase by fitting a cosine to the response after removing the first cycle (to eliminate the effect of the grating appearance). When presented in *Figure 5E*, phase is shown between $-2\pi$ and $\pi$ to facilitate the separation of PD and ND responses for moving grating starting at the two different phases.

## Statistics

To determine statistically significant differences, the one-sided, unpaired Student's *t*-test was used for comparing groups (*Figures 1E*, *2C,* and *3D*). In *Figure 3D*, we controlled for the false discovery rate using the Benjamini and Hochberg procedure (*Benjamini and Hochberg, 1995*), with q = 0.075. We noted that data were approximately normally distributed in general, but no formal test was conducted. Regression analysis for *Figure 3E* and *Figure 3—figure supplement 1D*) was performed using MATLAB fit function, fitting a first-degree polynomial. No statistical methods were used to pre-determine sample sizes, however our sample sizes are similar to those reported in previous publications (*Bahl et al., 2015*; *Turner-Evans et al., 2017*; *Tuthill et al., 2014*). Data collection and analysis could not be performed blind to the conditions of the experiments.

## Data plotting conventions

All boxplots presented were plotted with these conventions: box represents upper and lower quartile range, line represents median, whiskers were omitted, and individual data points are overlaid on the box.

## T5 neuron models (*Figures 4*, *5* and *7*)
### Model membrane potential

The models that follows are largely based on the T4 model we previously introduced (*Gruntman et al., 2018*). We modeled the dynamics of T5 somatic membrane potential, $V(t)$, as a single-compartment conductance-based neuron

$$\tau \dot{V}(t) = -(V(t) - V_L) - (g_E^* + g_E(t))(V(t) - V_E) - (g_I^* + g_I(t))(V(t) - V_I) \tag{1}$$

where $\tau$ denotes the integration time constant of the model neuron and $V_E$, $V_I$, $V_L$ denote, respectively, the excitatory, inhibitory, and leak reversal potentials. $g_E^*$ denotes a baseline excitatory conductance that is stimulus-independent, from which a stimulus-dependent conductance could be added or removed ($g_E(t)$). To ensure the baseline membrane potential will not drift due to the stimulus-independent conductance $g_E^*$, we added a stimulus-independent inhibitory global conductance, $g_I^*$. Finally, $g_I(t)$ denotes a stimulus-dependent inhibitory conductance. To simplify the model, and based on our results from the T4 model (*Gruntman et al., 2018*), we assumed negligible integration on the membrane and set $\tau = 0$.

The model membrane potential (*Equation 1*) can thus be written as

$$V = \frac{V_L + (g_E^* + g_E)V_E + (g_I^* + g_I)V_I}{1 + g_E^* + g_I^* + g_E + g_I} \tag{2}$$

where, for simplicity, we omitted the time dependence in the conductances and membrane potential. When there is no input stimulus into the system, the baseline voltage is

$$V_b = \frac{V_L + g_E^* V_E + g_I^* V_I}{1 + g_E^* + g_I^*} \tag{3}$$

Note that in the presence of stimulus-independent conductances, the baseline voltage is not necessarily identical to the leak reversal potential. The difference between baseline voltage and leak reversal potential is

$$V_b - V_L = \frac{g_E^*(V_E - V_L) + g_I^*(V_I - V_L)}{1 + g_E^* + g_I^*} = (V_E - V_L)\frac{g_E^* - \alpha g_I^*}{1 + g_E^* + g_I^*}; \alpha = \frac{V_L - V_I}{V_E - V_L} \tag{4}$$

The membrane potential in *Equation 2* can thus be written as

$$V = \frac{V_b(1 + g_E^* + g_I^*) + g_E V_E + g_I V_I}{1 + g_E^* + g_I^* + g_E + g_I} \tag{5}$$

We can then compute the membrane potential change with respect to baseline as

$$V - V_b = \frac{(V_E - V_b)g_E + (V_I - V_b)g_I}{1 + g_E^* + g_I^* + g_E + g_I} = (V_E - V_b)\frac{g_E' - \alpha' g_I'}{1 + g_E' + g_I'} \tag{6}$$

where $g_E' = \frac{g_E}{1 + g_E^* + g_I^*}$, $g_I' = \frac{g_I}{1 + g_E^* + g_I^*}$ denote the conductances normalized to the stimulus-independent total conductance, and $\alpha' = \frac{V_b - V_I}{V_E - V_b}$.

Finally, we can define an integrated conductance $\Delta G = g_E' - \alpha' g_I'$ and a total conductance $\Sigma G = 1 + g_E' + g_I'$, so that the model response (*Equation 6*) can be written as

$$V - V_b = (V_E - V_b)\frac{\Delta G}{\Sigma G} \tag{7}$$

## T5 neuron model – EI version

In this model there is a stimulus-dependent excitatory conductance ($g_E = E$), a stimulus-dependent inhibitory conductance ($g_I = I$), and no stimulus-independent conductances ($g_I^* = g_E^* = 0$). With these choices, the baseline membrane potential (*Equation 3*) becomes $V_b = V_L$, and the membrane potential (*Equation 6*) becomes

$$V - V_L = (V_E - V_L)\frac{E - \alpha I}{1 + E + I} \tag{8}$$

as in our previously published model (*Gruntman et al., 2018*).

## T5 neuron model – E₊E₋ version (*Figures 6* and *7*)

We assume that both the stimulus-dependent added conductance (E₊) and the stimulus-dependent removed conductance (E₋) share the same reversal potential and therefore represent the total stimulus-dependent conductance in the model as $g_E = E_+ - E_-$.

We rectified $E_-$ to prevent negative conductance values, that is $E_-$ never exceeds the stimulus independent conductance $(g_E^*)$. The stimulus-dependent inhibitory conductance was assumed to be absent $(g_I = 0)$.

### Bound for E₋ conductance

With these choices, we can write *Equation 6* as

$$V - V_b = (V_E - V_b)\frac{E'_+ - E'_-}{1 + E'_+ - E'_-} \tag{9}$$

where

$$E'_{+/-} = \frac{E'_{+/-}}{1 + g_E^* + g_I^*} \tag{10}$$

Since $E_-$ is removed, we demand $E_- \leq g_E^*$. Therefore, $E_- \leq \frac{g_E^*}{1 + g_E^* + g_I^*} < 1$

Additionally, from *Equation 4* we know that

$$V_b \cong V_L \Rightarrow g_E^* \cong \alpha g_I^* \tag{11}$$

Leading to the following bound for the E₋ conductance

$$E'_- \leq \frac{g_E^*}{1 + g_E^* + g_E^*/\alpha} \tag{12}$$

## T5 neuron model – scaled EI version (*Figure 7*)

To generate this version of the model we used the dynamics of the EI model with the spatial and temporal parameters from the E₊E₋ model. The parameters of the E conductance were matched to the E₊ conductance, while the parameters of the I conductance were matched to the E₋ conductance. Due to the stimulus-independent conductances in the E₊E₋ model the amplitudes could not be directly matched. Therefore, we scaled the E and I conductance to match the numerator of the E₊E₋ model so that the depolarizing and hyperpolarizing conductances are equal.

## Model conductances, inputs, and optimization

### Model conductances

The dynamics of the excitatory and inhibitory conductances $(k(t); k = E, E_+, E_-, I)$, measured in units of leak conductance, are described by two first-order linear filters in series

$$\begin{aligned} \tau_k^R \dot{h}_k(t) &= -h_k(t) + I_k^{stim}(t) \\ \tau_k^D \dot{k}(t) &= -k(t) + h_k(t) \end{aligned} \tag{13}$$

where $(\tau_k^R; \tau_k^D; k = E, E_+, E_-, I)$ denote the rise and decay time constants of the conductances, and $I_k^{stim}(t)$ is determined by the spatial receptive fields of the conductances and the spatiotemporal profile of the visual stimulus.

### Model inputs

The input to the conductance equations

$$I_k^{stim}(t) = \Sigma_x R_k(x)\,S(x,t),$$

is determined by the spatiotemporal profile of the visual stimulus $S(x,t)$ and the spatial receptive field of the conductance ($R_k(x)$; $k = E, E_+, E_-, I$). The integer index $x$ runs over locations along the ND-PD axis of a cell. This spatial discretization is dictated by the smallest size of the bars used in the experiment (width 1). The arbitrary reference $x = 0$ denotes the empirically determined location in the cell receptive field where a flashed bar elicited the strongest depolarization. The stimulus $S(x,t)$ can assume two possible values, 0 or 1, denoting respectively the absence or presence of an OFF bar of width one at the corresponding spatial location and time. We are not explicitly considering the height of the bar, hence our model stimuli replicate the one-dimensional equivalent of the stimuli used in the experiments.

## Receptive fields

We modeled the spatial receptive field of the excitatory and inhibitory conductances as a Gaussian profile

$$R_k(x) = A_k e^{-\dfrac{(x - \mu_k)^2}{2\sigma_k^2}}$$

centered at location $\mu_k$, with amplitude $A_k$ and width $\sigma_k$ ($k = E, E_+, E_-, I$).

## Model optimization

We numerically integrated the model dynamics with a 4th-order Runge-Kutta integration scheme. We obtained model parameters by numerically minimizing the squared error between the membrane potential dynamics of the model and the measured membrane potential of T5 cells in response to flashed bars of width 2. A constrained minimization procedure was performed 1000 times for each cell, starting from random uniform initialization of the parameter values within specified bounds (*Supplementary file 1*). Predictions for the remaining sets of stimuli from the top 1% of models (10/1000, based only on the magnitude of the error for the width 2 bar flash stimuli) were then compared with the measured responses (*Figures 4* and *5*). All simulations were performed in Matlab. Code and data are available at https://github.com/reiserlab/T5ConductanceModel (copy archived at https://github.com/elifesciences-publications/T5ConductanceModel).

## Acknowledgements

We thank A Nern for providing the driver line and E Rogers for help with fly husbandry. We are also grateful to J E Fitzgerald, K D Longden, and A Zhao for comments on the manuscript. This project was supported by the Howard Hughes Medical Institute.

## Additional information

### Funding

| Funder | Author |
| --- | --- |
| Howard Hughes Medical Institute | Michael B Reiser<br>Sandro Romani |

The funders had no role in study design, data collection and interpretation, or the decision to submit the work for publication.

### Author contributions

Eyal Gruntman, Conceptualization, Software, Formal analysis, Investigation, Visualization, Methodology; Sandro Romani, Conceptualization, Software, Formal analysis, Methodology; Michael B Reiser, Conceptualization, Supervision, Funding acquisition

Author ORCIDs
Eyal Gruntman (iD) https://orcid.org/0000-0003-1383-7347
Sandro Romani (iD) https://orcid.org/0000-0002-4727-4207
Michael B Reiser (iD) https://orcid.org/0000-0002-4108-4517

Decision letter and Author response
Decision letter https://doi.org/10.7554/eLife.50706.sa1
Author response https://doi.org/10.7554/eLife.50706.sa2

## Additional files

### Supplementary files

• Supplementary file 1. Excitation-Inhibition model parameters (related to *Figures 4*, *5* and *7*).

• Supplementary file 2. Excitation-Removal of excitation model parameters (related to *Figures 6* and *7*).

• Transparent reporting form

### Data availability

All data collected in this study that are summarized in the figures have been made available on Fig-Share, at doi: 10.25378/janelia.c.4771805.v1. Figure plotting code has been made available together with these data. The code used to simulate the T5 models and to generate the model data has been made available on https://github.com/reiserlab/T5ConductanceModel.git (copy archived at https://github.com/elifesciences-publications/T5ConductanceModel).

The following dataset was generated:

| Author(s) | Year | Dataset title | Dataset URL | Database and Identifier |
| --- | --- | --- | --- | --- |
| Reiser M, Gruntman E, Romani S | 2019 | The computation of directional selectivity in the Drosophila OFF motion pathway | https://doi.org/10.25378/janelia.c.4771805.v1 | figshare, 10.25378/janelia.c.4771805.v1 |

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
