## [Decision Letter]

**Acceptance summary:**

*Drosophila* has emerged as a powerful organism to study the basic principles of motion vision. The T4 and T5 system, with its ability to detect motion in all four cardinal directions is a paradigm that you explore in detail in your paper. Your previous work on T4 cells was pioneering in the sense that it clarified how motion could be detected using a more complex system that a simple Reichardt detector. However, T5s were more difficult to investigate, on the one hand because of a lack of a good connectomics data of this region of the lobula, but also because it was less clear how activation and potential inhibition were to result in motion detection. Your paper achieves a deep understanding of the process, and the attached modeling will be useful to test hypotheses that remain and will be addressed when more is known about the connections to T5 neurons.

We wish to congratulate you for this elegant work that completes the story of primary motion detection in flies and that will serve as a paradigm for other systems.

**Decision letter after peer review:**

Thank you for submitting your article "The computation of directional selectivity in the *Drosophila* OFF motion pathway" for consideration by *eLife*. Your article has been reviewed by Ronald Calabrese as the Senior Editor, a Reviewing Editor, and three reviewers. The following individuals involved in review of your submission have agreed to reveal their identity: Alon Poleg-Polsky (Reviewer #2).

The reviewers have discussed the reviews with one another and I have drafted this decision to help you prepare a revised submission.

The reviewers think that the intracellular recordings that you report are very useful for understanding how T4 and T5 become direction-sensitive. The fact that T4 and T5 appear to function is similar ways in spite of very different connectivity is of significant interest, although it would be important in a revised version to explain why this is important.

The reviewers also appreciated that you spent time investigating your model and to compare it with the T4 data.

Therefore, we do not ask for any new experiments, but we would like you to address a number of points about the modeling:

- Using only the peak of the Vm response as a measure of response strength for direction-selectivity might not be sufficient. It is straightforward to measure the integral of the responses.

- The modeling makes claims that are in conflict with published data and you should mention and discuss this, especially in the optimality portion. In general, although the reviewers do appreciate that you evaluate the performance of your model and its relationship to other data in the field, the analysis of these other results is only made in relation to your model: If your model disagrees with other data, this should be noted and discussed.

- You seem to cast the nonlinear aspect of the Vm response that you observe as the only origin of direction selectivity of T5, which does not appear to be a conclusion you can draw.

- You do not really know whether the hyperpolarization is caused by inhibition. In fact, you use your model to check whether inhibition rather than reduction of the excitatory signal is critical to the direction selectivity computation. In other words, you seem to imply from the nonlinear suppression of Vm towards baseline that the hyperpolarization must come from a mechanism with increased conductance, which might exclude a release-from-excitation mechanism. This could be a strong conclusion of the paper, especially if you highlight that all the inputs to T5 are cholinergic.

If a simple model works just fine with fewer parameters and without making some arbitrary assumptions such as having the chloride reversal potential being a free parameter, why not use it throughout the paper?

Therefore, in revising your model being more careful and comparing it with published data will help make the paper stronger.

Reviewer #1:

Gruntman et al., used electrophysiological measurements of membrane potential in the neuron T5, combined with visual apparent motion stimuli, to investigate and model direction-selective (DS) signals in that class of neurons. This work follows closely a similar previous paper investigating T4, and the results are quite similar. This paper goes into some more depth investigating properties of the fitted model. The quality of the experimental work and analysis is high and the conclusions will be of wide interest to those studying the mechanics of neural computations and direction-selectivity.

I found a few issues that could be profitably addressed. Some of these points are lengthy, but that is an effort on my part to be clear and to make sure we are not speaking at crossed purposes.

1) To make the case that a signal is DS, one must have a measure of the strength of the signal. This is straightforward with spiking neurons (e.g., spike rate), but far from obvious with analog signals like voltage or calcium. To assess the direction-selectivity of neural responses, the authors compare the maximum value of the Vm reached during the timecourse of the stimulus (Figure 3 top row, Figure 4E). This is an odd nonlinear metric to use to asses direction-selectivity. For one, there are no known or proposed biophysical mechanisms that I am aware of that could perform this operation. For another, the actual nonlinear method used to assess direction-selectivity can be critical. I have created an example using Matlab code in which two linear temporal filters are applied at two points in space and the results summed. When direction-selectivity is assessed by the maximum response, the model has one PD, but when it is assessed by the integrated squared response (an alternative method that could plausibly be biophysically implemented; also equivalent to the motion energy model), the PD of that same model is the opposite direction. Thus, it can matter a great deal how one assesses the strength of a response. Based on the traces in this paper, I don't think this data will have such perverse properties, but, given that it's possible, the authors must address why the maximum value is a reasonable way to assess response strength. Note that I am not arguing against using the maximum when comparing true responses to linear predictions, just in assessing the direction-selectivity of a response or a model.

Related to this, later in the manuscript, the authors assess response strength by the f1 amplitude of the response to a periodic stimulus (Figure 5 and Figure 6). This assessment of response strength is no longer the response maximum but related to the mean squared response. It might be helpful to readers to use a consistent method to evaluate response strengths.

2) There seemed to be some issues in the motivation and analysis of the model. I have listed the most important ones below.

- Figure 6B. The TF and SF axes have real units for this model in Hz and cycles/degree, and it would be most helpful to show them, rather than these normalized units.

- Figure 6BC. There exists a precise definition for TF-tuning, namely that the surface be separable into the product of functions of TF alone and of SF alone (see Priebe, Lisberger and Movshon, (2006)). That definition should probably be used to make claims about the presence of TF-tuning.

- Closely related to the point above: an experimental measurement exists for mean T5 calcium responses as a function of SF and TF in Creamer et al., (2018). It shows that T5 units on average show TF-tuning; it does not show the ND lobe at low SF that exists in this figure. The authors focused at this point in the text on how their DSI surface can permit a model for velocity-tuned behavior simpler than that proposed in Creamer et al., 2018 but they do not mention that their model does not match existing calcium measurements in T5 in response to sinewaves of different TF and SF.

- Figure 6C. The DSI surface indeed looks velocity tuned; since the DSI involves the quantity (r(PD) – r(ND)), doesn't this mean that the model implies that HS and VS should be velocity-tuned? The opponency in the input to these LPTCs is equivalent to this computation of r(PD) – r(ND). There is abundant evidence that HS and VS are TF-tuned.

- Optimality in 6DE. Here, as I read it, the authors chose a single TF and SF sinewave, and then swept two model parameters: the ratio timescales for excitation and inhibition (D) and the normalized distance between input filters (E). Heuristically, the speed to which a detector is most sensitively DS is roughly δ X/δ T, where δ T is related to the ratio \δ here and δ X is related to \Chi. I would therefore expect that the optimal \δ should be exquisitely sensitive to the TF and SF combination of the sinusoid, so for another (SF,TF) pairing, say a really fast or extraordinarily slow sinewave, the fitted model parameters would be far from the \δ corresponding to peak DSI. The same arguments apply to the sweep of \Chi – the value of \Chi with the maximum DSI seems likely to depend strongly on the speed of the stimulus. In both cases, I don't really understand how the system could be said to be optimal, since it's only optimal for a specific stimulus, near where the parameters were fit. If the optimality argument depends strongly on the stimulus chosen to assess it, then the claims about optimality should probably be removed, since it's not clear how to choose that stimulus (or distribution of stimuli).

– In subsection “Model analysis reveals near-optimal tuning for directionally selective responses to grating stimuli”, the authors suggest that the f1 components of T4 and T5 cells should sum to create the steady state response in LPTCs. If the voltage signal were transferred linearly to LPTCs, then summing over sinewaves with different phases across space would give zero. If before summing, a quadratic nonlinearity is applied to each voltage, like a motion energy model, then this statement becomes true. The assumptions about nonlinearities should be laid out clearly. Similarly, modeling T4 and T5 output (i.e., the DSI surface in 6C) by their f1 amplitude makes a strong assumption about the nonlinearity involved in transferring their signals to downstream cells.

3) The relationship to calcium imaging experiments may deserve a bit more explanation. As I read the paper, one possible outcome of these experiments was that all voltage traces could have been found to be linear – that the Vm was a perfect linear function of the input contrast. In that case, the cells would still be direction-selective as long as the linear filter was oriented in space-time and there was some kind of nonlinearity, like an expansive nonlinearity, applied to the voltage, perhaps in the transformation into calcium. Such a model has been suggested by Leong et al., Wienecke et al., and Badwan et al., (with caveats there; that paper would seem to be consistent with the small Vm nonlinearities you found here). This manuscript focuses on the smallish nonlinearity in the Vm caused by inhibitory conductances shunting excitatory currents, but in its modeling section, it focuses on the purely linear model for Vm produced by fitting a model. If the voltage were purely linear, it would still permit PD enhancement and ND suppression in the calcium if there were a nonlinear relationship between V and Ca, for instance a quadratic relationship. (Alternatively, these authors hint, the nonlinearity could be in the calcium indicator as well or instead.) If the summary above is correct, it would be useful to have some approximation of it in the paper; as it stands, there are some very confusing statements and no clear explanation in the Discussion section of how these results fit with the calcium measurements made elsewhere.

Subsection “T5 neurons generate directional selectivity using only ND suppression”: "suppression of ND motion is the only mechanism through which DS is generated in T5 neurons"? This statement is far too strong. At the very least, it's the only mechanism of direction-selectivity found in Vm using this set of stimuli; other mechanisms could exist (and have been suggested) in the V-Ca transformation. Furthermore, even if no nonlinearities had been found in Vm, the cell could be DS, depending on the subsequent nonlinearity that acts on the Vm. We don't know whether the voltage-calcium transformation does anything unusual, as proposed as possibilities in both Wienecke and Badwan.

Discussion section: "significance of calcium transformation remains unclear." Do the authors believe that the nonlinearities they observed in Vm could account for the full direction-selectivity of T5? It must also depend on any subsequent nonlinearities acting on Vm – their modeling makes a specific choice of nonlinearity acting on the Vm. Haag et al., (2017) used very similar stimuli to obtain PD enhancement, while Fisher, (2015) used a bit different stimulus and Salazar-Gatzimas, (2016) used a very different stimulus. These calcium imaging experiments are not inconsistent with this data; they just suggest that the PD enhancement occurs after the Vm computation, while ND suppression is already present in Vm.

A more concrete discussion of how cellular quantities downstream of Vm might transform it nonlinearly would help to address the other data in the literature, which is primarily calcium imaging.

Reviewer #2:

This manuscript is a continuation of the work published by the same authors last year (Gruntman et al., 2018) that uses patch-clamp recordings to investigate direction selectivity (DS) mechanisms in *Drosophila*. The focus of the current project is on processing in T5 cells. The authors nicely summarized their work in the first paragraph of the Discussion section, I wouldn't repeat it here.

Overall, while not as innovative as their previous work, I find this project to be a solid investigation of the relevant visual processing in T5 cells. I like the progression from simple to more complex in terms of visual stimulation and the level of analysis. The authors expand upon their experimental findings with two computational models that allow them to compare the different visual paradigms and establish the main mechanism of DS computation in these cells.

I do however have a number of concerns regarding the current version of the manuscript.

1) The authors decided to analyze most of the data using bar widths and not in terms of visual angles. Although the conversion (1 pixel=~2 degrees of visual space) is mentioned in methods and the result sections, I felt that the constant conversion required from the reader imposes an unnecessary mental penalty. Moreover, the usage of bar width comparisons is problematic for diagonal stimuli (45, 135, 225 and 315 degrees), which have a different angular resolution due to the restrictions of the display apparatus. The authors are aware of this fact and for some of their analysis use only cells with the preferred direction that follows the cardinal directions (Figure 5).

Because DS is highly affected by stimulus width (Figure 1, Figure 3), cells that are tuned to cardinal and diagonal directions should not be grouped together – unless the authors can provide evidence for the validity of their approach.

2) I found that the metrics the authors used to cross-validate the model are not intuitive and do not provide a clear understanding of the strengths and the weaknesses of the model. For example, Figure 4 (A conductance-based model quantitatively predicts directionally selective responses) compare the original traces with the output of the model. The example cell in panel B shows an interesting trend – the model was fitted to stimuli with bar width 2, and then it overshoots ND responses for bar width 1 and PD responses for bar width 4. Is that a fluke or a trend? I can't tell from panel C or from the supplementary figure. Perhaps a more useful analysis would be to plot the predictions of the full model against a linear summation of individual bar responses and against the simpler model the authors use to fit responses to drifting gratings.

*Reviewer #3:*

This is a very well-constructed study with some excellent data, but there is a reason for that. It's largely already been published! Gruntman et al., 2018 performed essentially the same experiments, analysis and modeling in the T4 ON pathway, and now they are trying to publish an almost identical paper on the T5 OFF pathway. The DS computation turns out to be virtually identical in these two cell types.

While this is a solid piece of work, I am not sure it reaches the "novelty" bar for *eLife*. I will defer to the editor for that determination, but I am just giving my opinion on the matter.

---

## [Author Response]

The reviewers think that the intracellular recordings that you report are very useful for understanding how T4 and T5 become direction-sensitive. The fact that T4 and T5 appear to function is similar ways in spite of very different connectivity is of significant interest, although it would be important in a revised version to explain why this is important.The reviewers also appreciated that you spent time investigating your model and to compare it with the T4 data.Therefore, we do not ask for any new experiments, but we would like you to address a number of points about the modeling:- Using only the peak of the Vm response as a measure of response strength for direction-selectivity might not be sufficient. It is straightforward to measure the integral of the responses.

We have generated a version of Figure 3 using the mean response instead of the peak Vm. Figure 3 presents the T5 responses to bar pairs, and is the main result showing no PD enhancement (based on our analysis of peak responses). This new analysis did not change any of the conclusions drawn from these data and is now included in the Results section accompanying Figure 3 and as Figure 3—figure supplement 1. Further comments on this point are below in the response to reviewer #1.

- The modeling makes claims that are in conflict with published data and you should mention and discuss this, especially in the optimality portion. In general, although the reviewers do appreciate that you evaluate the performance of your model and its relationship to other data in the field, the analysis of these other results is only made in relation to your model: If your model disagrees with other data, this should be noted and discussed.

In the revised manuscript, we replaced the ‘optimality’ modelling analysis with an evaluation of an important, alternative model, and greatly expanded the discussion around discrepancies between our model and other results in the field (including new Figure 8).

- You seem to cast the nonlinear aspect of the Vm response that you observe as the only origin of direction selectivity of T5, which does not appear to be a conclusion you can draw.

In the revised manuscript, we further emphasize that linear models are insufficient to explain T5 responses. In the expanded discussion we now propose a unifying framework (Figure 8) that allows for an additional downstream nonlinearity.

- You do not really know whether the hyperpolarization is caused by inhibition. In fact, you use your model to check whether inhibition rather than reduction of the excitatory signal is critical to the direction selectivity computation. In other words, you seem to imply from the nonlinear suppression of Vm towards baseline that the hyperpolarization must come from a mechanism with increased conductance, which might exclude a release-from-excitation mechanism. This could be a strong conclusion of the paper, especially if you highlight that all the inputs to T5 are cholinergic.

We thank the reviewers for this astute suggestion. We have pursued this comparison more rigorously and present the results in new Figure 6. To our surprise, a release-from-excitation model cannot explain the directional selectivity of T5 neurons. Intuition for this discrepancy is provided in new Figure 7. Also, we have adopted a much more precise use of these terms such that we now refer to our measurements as showing depolarization or hyperpolarization (not excitation or inhibition) and just use E and I when we explore models and discuss the circuit implementation.

If a simple model works just fine with fewer parameters and without making some arbitrary assumptions such as having the chloride reversal potential being a free parameter, why not use it throughout the paper?

We very much appreciate this suggestion. Upon closer examination we were able to improve our optimization procedure and simplify the model, such that the model now presented is even simpler than our previous T4 model.

Therefore, in revising your model being more careful and comparing it with published data will help make the paper stronger.Reviewer #1:Gruntman et al., used electrophysiological measurements of membrane potential in the neuron T5, combined with visual apparent motion stimuli, to investigate and model direction-selective (DS) signals in that class of neurons. This work follows closely a similar previous paper investigating T4, and the results are quite similar. This paper goes into some more depth investigating properties of the fitted model. The quality of the experimental work and analysis is high and the conclusions will be of wide interest to those studying the mechanics of neural computations and direction-selectivity.

We thank the reviewer for this very positive assessment of our work, and for the very thoughtful and detailed comments.

I found a few issues that could be profitably addressed. Some of these points are lengthy, but that is an effort on my part to be clear and to make sure we are not speaking at crossed purposes.1) To make the case that a signal is DS, one must have a measure of the strength of the signal. This is straightforward with spiking neurons (e.g., spike rate), but far from obvious with analog signals like voltage or calcium. To assess the direction-selectivity of neural responses, the authors compare the maximum value of the Vm reached during the timecourse of the stimulus (Figure 3 top row, Figure 4E). This is an odd nonlinear metric to use to asses direction-selectivity. For one, there are no known or proposed biophysical mechanisms that I am aware of that could perform this operation. For another, the actual nonlinear method used to assess direction-selectivity can be critical. I have created an example using Matlab code in which two linear temporal filters are applied at two points in space and the results summed. When direction-selectivity is assessed by the maximum response, the model has one PD, but when it is assessed by the integrated squared response (an alternative method that could plausibly be biophysically implemented; also equivalent to the motion energy model), the PD of that same model is the opposite direction. Thus, it can matter a great deal how one assesses the strength of a response. Based on the traces in this paper, I don't think this data will have such perverse properties, but, given that it's possible, the authors must address why the maximum value is a reasonable way to assess response strength. Note that I am not arguing against using the maximum when comparing true responses to linear predictions, just in assessing the direction-selectivity of a response or a model.

As the reviewer notes, this could be a legitimate concern, but as we show it is not a major factor here. Because there are no definitive measurements relating synaptic transmission between T5 and downstream partners, any choice of a non-linearity will be somewhat arbitrary. We expect that there should be a monotonic relationship between T5 membrane potential and the post-synaptic depolarization of a downstream neuron, so we think the maximum of the T5 response is as reasonable as any other measure.

We expected that a mean of T5 responses would be well correlated to the max, because in nearly all instances, the responses for which we evaluated with the maximum value have a clear single peak. To illustrate this point, we have generated a new version of Figure 3 using the response mean instead of response peak. We specifically focused on Figure 3 because our previous conclusions relied on the peak calculation. This new analysis based on the mean did not change any of the conclusions drawn from these data. Figure 3—figure supplement 1 now includes the mean results.

Related to this, later in the manuscript, the authors assess response strength by the f1 amplitude of the response to a periodic stimulus (Figure 5 and Figure 6). This assessment of response strength is no longer the response maximum, but related to the mean squared response. It might be helpful to readers to use a consistent method to evaluate response strengths.

Since the response to a grating stimulus is periodic, and qualitatively different from responses to a flash or a moving bar, it would not be appropriate to summarize these data with a response peak. Similarly, it would not be sensible to extract an f1 amplitude from single-peaked responses. Respectfully, we suggest that a peak for single-peaked data (or equivalently, the mean, as we have shown) and an f1 amplitude for periodic responses are conventional, and reasonable ways to treat these data.

2) There seemed to be some issues in the motivation and analysis of the model. I have listed the most important ones below.- Figure 6B. The TF and SF axes have real units for this model in Hz and cycles/degree, and it would be most helpful to show them, rather than these normalized units.- Figure 6BC. There exists a precise definition for TF-tuning, namely that the surface be separable into the product of functions of TF alone and of SF alone (see Priebe, Lisberger and Movshon, (2006)). That definition should probably be used to make claims about the presence of TF-tuning.- Closely related to the point above: an experimental measurement exists for mean T5 calcium responses as a function of SF and TF in Creamer et al., (2018). It shows that T5 units on average show TF-tuning; it does not show the ND lobe at low SF that exists in this figure. The authors focused at this point in the text on how their DSI surface can permit a model for velocity-tuned behavior simpler than that proposed in Creamer et al.; but they do not mention that their model does not match existing calcium measurements in T5 in response to sinewaves of different TF and SF.- Figure 6C. The DSI surface indeed looks velocity tuned; since the DSI involves the quantity (r(PD) – r(ND)), doesn't this mean that the model implies that HS and VS should be velocity-tuned? The opponency in the input to these LPTCs is equivalent to this computation of r(PD) – r(ND). There is abundant evidence that HS and VS are TF-tuned.- Optimality in 6DE. Here, as I read it, the authors chose a single TF and SF sinewave, and then swept two model parameters: the ratio timescales for excitation and inhibition (D) and the normalized distance between input filters (E). Heuristically, the speed to which a detector is most sensitively DS is roughly δ X/δ T, where δ T is related to the ratio \δ here and δ X is related to \Chi. I would therefore expect that the optimal \δ should be exquisitely sensitive to the TF and SF combination of the sinusoid, so for another (SF,TF) pairing, say a really fast or extraordinarily slow sinewave, the fitted model parameters would be far from the \δ corresponding to peak DSI. The same arguments apply to the sweep of \Chi – the value of \Chi with the maximum DSI seems likely to depend strongly on the speed of the stimulus. In both cases, I don't really understand how the system could be said to be optimal, since it's only optimal for a specific stimulus, near where the parameters were fit. If the optimality argument depends strongly on the stimulus chosen to assess it, then the claims about optimality should probably be removed, since it's not clear how to choose that stimulus (or distribution of stimuli).- In subsection “Model analysis reveals near-optimal tuning for directionally selective responses to grating stimuli”, the authors suggest that the f1 components of T4 and T5 cells should sum to create the steady state response in LPTCs. If the voltage signal were transferred linearly to LPTCs, then summing over sinewaves with different phases across space would give zero. If before summing, a quadratic nonlinearity is applied to each voltage, like a motion energy model, then this statement becomes true. The assumptions about nonlinearities should be laid out clearly. Similarly, modeling T4 and T5 output (i.e., the DSI surface in 6C) by their f1 amplitude makes a strong assumption about the nonlinearity involved in transferring their signals to downstream cells.

We thank the review for these very thoughtful comments. Upon further reflection, we agree that this analysis of periodic stimulus responses was insufficiently constrained by our current experimental measurements. We therefore redirected our modelling effort towards answering more specific issues raised by our data. Consequently, the issues raised about interpretations of optimality no longer apply.

3) The relationship to calcium imaging experiments may deserve a bit more explanation. As I read the paper, one possible outcome of these experiments was that all voltage traces could have been found to be linear – that the Vm was a perfect linear function of the input contrast. In that case, the cells would still be direction-selective as long as the linear filter was oriented in space-time and there was some kind of nonlinearity, like an expansive nonlinearity, applied to the voltage, perhaps in the transformation into calcium. Such a model has been suggested by Leong et al., Wienecke et al., and Badwan et al., (with caveats there; that paper would seem to be consistent with the small Vm nonlinearities you found here). This manuscript focuses on the smallish nonlinearity in the Vm caused by inhibitory conductances shunting excitatory currents, but in its modeling section, it focuses on the purely linear model for Vm produced by fitting a model. If the voltage were purely linear, it would still permit PD enhancement and ND suppression in the calcium if there were a nonlinear relationship between V and Ca, for instance a quadratic relationship. (Alternatively, these authors hint, the nonlinearity could be in the calcium indicator as well or instead.) If the summary above is correct, it would be useful to have some approximation of it in the paper; as it stands, there are some very confusing statements and no clear explanation in the Discussion section of how these results fit with the calcium measurements made elsewhere.

We thank the reviewer for pointing out this potential interpretation, which has now been clarified in the revised manuscript. To be clear, our view is that the primary mechanism of directional selectivity is the non-linear integration of offset E and I inputs. We do not oppose the suggestion that a secondary nonlinearity due to the voltage to calcium transformation may also contribute and have proposed a unifying framework in Figure 8. Where we depart from the reviewer’s suggested interpretation is that we do not believe the nonlinearity we directly observe in the voltage signal is “smallish” or negligible. In our previously submitted manuscript, we had focused the linearity comparison to fairly weak stimuli (bar pairs, Figure 3). To clarify the contribution of nonlinear integration, we have added the linear superposition as a comparison for moving bar responses in Figure 4. As is readily apparent, especially in the case of wider bar stimuli, the T5 response is very poorly approximated by the linear sum, while the EI model predicts it very well. Furthermore, our new comparative analysis between the EI and E_+_E_-_ model (Figure 7) explicitly shows the importance of the dynamic non-linearity for generating DS responses. The text accompanying these figures has been expanded to explain these points.

Subsection “T5 neurons generate directional selectivity using only ND suppression”: "suppression of ND motion is the only mechanism through which DS is generated in T5 neurons"? This statement is far too strong. At the very least, it's the only mechanism of direction-selectivity found in Vm using this set of stimuli; other mechanisms could exist (and have been suggested) in the V-Ca transformation. Furthermore, even if no nonlinearities had been found in Vm, the cell could be DS, depending on the subsequent nonlinearity that acts on the Vm. We don't know whether the voltage-calcium transformation does anything unusual, as proposed as possibilities in both Wienecke and Badwan.

We agree the previous statement was too strong and have revised it.

Discussion section: "significance of calcium transformation remains unclear." Do the authors believe that the nonlinearities they observed in Vm could account for the full direction-selectivity of T5? It must also depend on any subsequent nonlinearities acting on Vm – their modeling makes a specific choice of nonlinearity acting on the Vm. Haag et al., (2017) used very similar stimuli to obtain PD enhancement, while Fisher, (2015) used a bit different stimulus and Salazar-Gatzimas, (2016) used a very different stimulus. These calcium imaging experiments are not inconsistent with this data; they just suggest that the PD enhancement occurs after the Vm computation, while ND suppression is already present in Vm.A more concrete discussion of how cellular quantities downstream of Vm might transform it nonlinearly would help to address the other data in the literature, which is primarily calcium imaging.

The sentence on in the Discussion section has been removed. The potential significance of the voltage-to-calcium transformation now gets a much fuller treatment in the discussion (alongside Figure 8).

Reviewer #2:This manuscript is a continuation of the work published by the same authors last year (Gruntman et al., 2018) that uses patch-clamp recordings to investigate direction selectivity (DS) mechanisms in Drosophila. The focus of the current project is on processing in T5 cells. The authors nicely summarized their work in the first paragraph of the Discussion section, I wouldn't repeat it here.Overall, while not as innovative as their previous work, I find this project to be a solid investigation of the relevant visual processing in T5 cells. I like the progression from simple to more complex in terms of visual stimulation and the level of analysis. The authors expand upon their experimental findings with two computational models that allow them to compare the different visual paradigms and establish the main mechanism of DS computation in these cells.

We thank the reviewer for this positive assessment of our work, and for the thoughtful and detailed comments.

I do however have a number of concerns regarding the current version of the manuscript.1) The authors decided to analyze most of the data using bar widths and not in terms of visual angles. Although the conversion (1 pixel=~2 degrees of visual space) is mentioned in methods and the result sections, I felt that the constant conversion required from the reader imposes an unnecessary mental penalty. Moreover, the usage of bar width comparisons is problematic for diagonal stimuli (45, 135, 225 and 315 degrees), which have a different angular resolution due to the restrictions of the display apparatus. The authors are aware of this fact and for some of their analysis use only cells with the preferred direction that follows the cardinal directions (Figure 5).

The reviewer raises a fair point. Since we are measuring responses in a live animal (with a non-flat retina), and since our arena is cylindrical (maintaining a constant visual-angle-to-width ratio only in the horizontal plane), the conversion between pixels and visual angle can only be estimated. As an extreme example, this cylindrical distortion causes a 2-pixel-wide bar in the center to correspond to a 4.5°-wide visual angle, while the same 2-pixel-wide bar at the edge of our arena would correspond to ~3.34°. For this reason, and to maintain internal consistency across figures, we prefer to retain the pixel-based stimulus characterization as the primary description. However, to alleviate the need for frequent conversions we have added the approximate angular equivalent for (cardinal) bar widths to Figure 2, Figure 3, and Figure 4, and to the figure legend for Figure 1. To address the concern about diagonal versus cardinal stimulus presentations, we now include Figure 2—figure supplement 1 that shows the receptive field mapping calculated separately for diagonal and cardinal cells, corrected for approximate stimulus width as visual angles.

Because DS is highly affected by stimulus width (Figure 1, Figure 3), cells that are tuned to cardinal and diagonal directions should not be grouped together – unless the authors can provide evidence for the validity of their approach.

We have added Figure 2—figure supplement 1 that shows the receptive field mapping calculated separately for diagonal and cardinal cells. The figure shows that while there are minor differences between the 2 groups when viewed by position, these differences are negligible when corrected for the approximate visual angle.

2) I found that the metrics the authors used to cross-validate the model are not intuitive and do not provide a clear understanding of the strengths and the weaknesses of the model. For example, Figure 4 (A conductance-based model quantitatively predicts directionally selective responses) compare the original traces with the output of the model. The example cell in panel B shows an interesting trend – the model was fitted to stimuli with bar width 2, and then it overshoots ND responses for bar width 1 and PD responses for bar width 4. Is that a fluke or a trend? I can't tell from panel C or from the supplementary figure. Perhaps a more useful analysis would be to plot the predictions of the full model against a linear summation of individual bar responses and against the simpler model the authors use to fit responses to drifting gratings.

We thank the reviewer for this helpful suggestion. The comparison in the original manuscript emphasized that there were no systematic discrepancies, but without an intuitive reference. We now have added the linear estimate (flash response superposition) to Figure 4, so that our model now benefits from an important benchmark, and gives a concrete example of what a systematic prediction error looks like.